# Atlas of *Plasmodium falciparum* intraerythrocytic development using expansion microscopy

**Benjamin Liffner[1][†], Ana Karla Cepeda Diaz[2,3][†], James Blauwkamp[1], David Anaguano[4,5], Sonja Frolich[6,7], Vasant Muralidharan[4,5], Danny W Wilson[6,8,9], Jeffrey D Dvorin[3,10], Sabrina Absalon[1]\***

[1]Department of Pharmacology and Toxicology, Indiana University School of Medicine, Indianapolis, United States; [2]Biological and Biomedical Sciences, Harvard Medical School, Boston, United States; [3]Division of Infectious Diseases, Boston Children's Hospital, Boston, United States; [4]Center for Tropical and Emerging Global Diseases, University of Georgia, Athens, United States; [5]Department of Cellular Biology, Franklin College of Arts and Sciences, University of Georgia, Athens, United States; [6]Research Centre for Infectious Diseases, School of Biological Sciences, University of Adelaide, Adelaide, Australia; [7]Institute for Photonics and Advanced Sensing, University of Adelaide, Adelaide, Australia; [8]Institute for Photonics and Advanced Sensing, University of Adelaide, Adelaide, Australia; [9]Burnet Institute, 85 Commercial Road, Melbourne, Australia; [10]Department of Pediatrics, Harvard Medical School, Boston, United States

**\*For correspondence:** sabsalon@iu.edu

[†]These authors contributed equally to this work

**Competing interest:** The authors declare that no competing interests exist.

**Abstract** Apicomplexan parasites exhibit tremendous diversity in much of their fundamental cell biology, but study of these organisms using light microscopy is often hindered by their small size. Ultrastructural expansion microscopy (U-ExM) is a microscopy preparation method that physically expands the sample by ~4.5×. Here, we apply U-ExM to the human malaria parasite *Plasmodium falciparum* during the asexual blood stage of its lifecycle to understand how this parasite is organized in three dimensions. Using a combination of dye-conjugated reagents and immunostaining, we have cataloged 13 different *P. falciparum* structures or organelles across the intraerythrocytic development of this parasite and made multiple observations about fundamental parasite cell biology. We describe that the outer centriolar plaque and its associated proteins anchor the nucleus to the parasite plasma membrane during mitosis. Furthermore, the rhoptries, Golgi, basal complex, and inner membrane complex, which form around this anchoring site while nuclei are still dividing, are concurrently segregated and maintain an association to the outer centriolar plaque until the start of segmentation. We also show that the mitochondrion and apicoplast undergo sequential fission events while maintaining an association with the outer centriolar plaque during cytokinesis. Collectively, this study represents the most detailed ultrastructural analysis of *P. falciparum* during its intraerythrocytic development to date and sheds light on multiple poorly understood aspects of its organelle biogenesis and fundamental cell biology.

## eLife assessment

This **important** study provides an unprecedented overview of the subcellular organization of proliferative blood-stage malaria parasites using expansion microscopy. The localization of multiple parasite organelles is comprehensively probed using three-dimensional super-resolution microscopy throughout the entire intraerythrocytic development cycle. This work provides a **compelling**

framework to investigate in future more deeply the unconventional cell biology of malaria-causing parasites.

## Introduction

The human malaria parasite *Plasmodium falciparum* has a complex lifecycle that involves both human and mosquito hosts. Of its many lifecycle stages, the asexual replication of *P. falciparum* inside human red blood cells (RBCs) is responsible for the clinical symptoms of malaria. This asexual blood stage starts when a merozoite invades a host RBC and transitions through several morphologies before forming approximately 30 new daughter merozoites (*Rudlaff et al., 2020*), which egress from their host cell and invade new RBCs (*Figure 1a*). Host RBCs are approximately 7–8 μm in diameter (*Kinnunen et al., 2011*) and contain dozens of parasites, each with their own sets of organelles and structures. The small size of *P. falciparum* and its organelles still poses a challenge to the study of many facets of *P. falciparum* cell biology, especially when immunostaining is required.

Expansion microscopy is a set of sample preparation techniques that isotropically increase the physical size of a microscopy sample (*Wassie et al., 2019*). While many expansion microscopy methods have been developed, ultrastructure expansion microscopy (U-ExM) (*Gambarotto et al., 2019*) was the first used in *Plasmodium* and since has been used in *P. falciparum* and across multiple apicomplexan parasites (*Bertiaux et al., 2021*; *Dave et al., 2022*; *Liffner and Absalon, 2021*; *Oliveira Souza et al., 2022*; *Rashpa and Brochet, 2022*; *Severo et al., 2022*). U-ExM results in the ~4.5-fold isotropic expansion of the sample and largely preserves its proteome, making it compatible with antibody staining and many fluorescent dyes (*Gambarotto et al., 2019*). The increase in physical sample size results in a dramatic increase in the ability to identify and distinguish different parasite structures. Thus, some structures that could previously only be investigated using electron microscopy can now be studied with the flexibility, scalability, and inexpensive nature of conventional light microscopy.

Application of U-ExM to *Plasmodium*, and other Apicomplexa, has already enhanced our understanding of parasite cell biology tremendously, resulting in the identification of new parasite structures and better characterization of the size and shape of others (*Bertiaux et al., 2021*; *Liffner and Absalon, 2021*; *Qian et al., 2022*; *Rashpa and Brochet, 2022*; *Simon et al., 2021*; *Tomasina et al., 2022*; *Tosetti et al., 2020*). Its significant impact on the field in such a short amount of time indicates U-ExM will be a technique heavily used in both *Plasmodium* and Apicomplexa more broadly for the foreseeable future. Considering this, we set out to image *P. falciparum* structures and organelles across the asexual blood stage of the lifecycle to serve as a reference for the expanding number of U-ExM users who study Apicomplexa and uncover previously invisible aspects of the cell biology of *P. falciparum*.

## Results

### U-ExM reveals multiple parasite structures without the use of antibodies

Dyes that are not antigen-specific are commonplace in light microscopy. N-hydroxysuccinimide (NHS) esters conjugated to dyes are amino-reactive and can be used for fluorescent labeling of protein density (*Nanda and Lorsch, 2014*). Similarly, BODIPY TR ceramide (BODIPY TRc) and other dye-conjugated fatty acids are commonly used for labeling lipids (*Marks et al., 2008*). Coupling U-ExM with these dyes has already revealed parasite structures for which specific antibodies did not exist (*Bertiaux et al., 2021*; *Liffner and Absalon, 2021*; *Simon et al., 2021*). While antibody-based labeling provides high specificity, this labeling lacks complexity and is limited to the specific protein or antigen that is being targeted. The more general stains increase the number of parasite features or organelles we can observe in the same sample without additional antibody markers. Therefore, these general stains allow for low-specificity but high-complexity imaging. An example of this principle is the use of uranyl acetate in electron microscopy to increase contrast by increasing the electron density of phosphate-rich structures in the cell (*Rudlaff et al., 2020*). While NHS esters and uranyl acetate can bring out similar features in the cell, they do not have the same reactivity and are therefore not equivalent stains. To better profile the subcellular organization of *P. falciparum* during the asexual blood

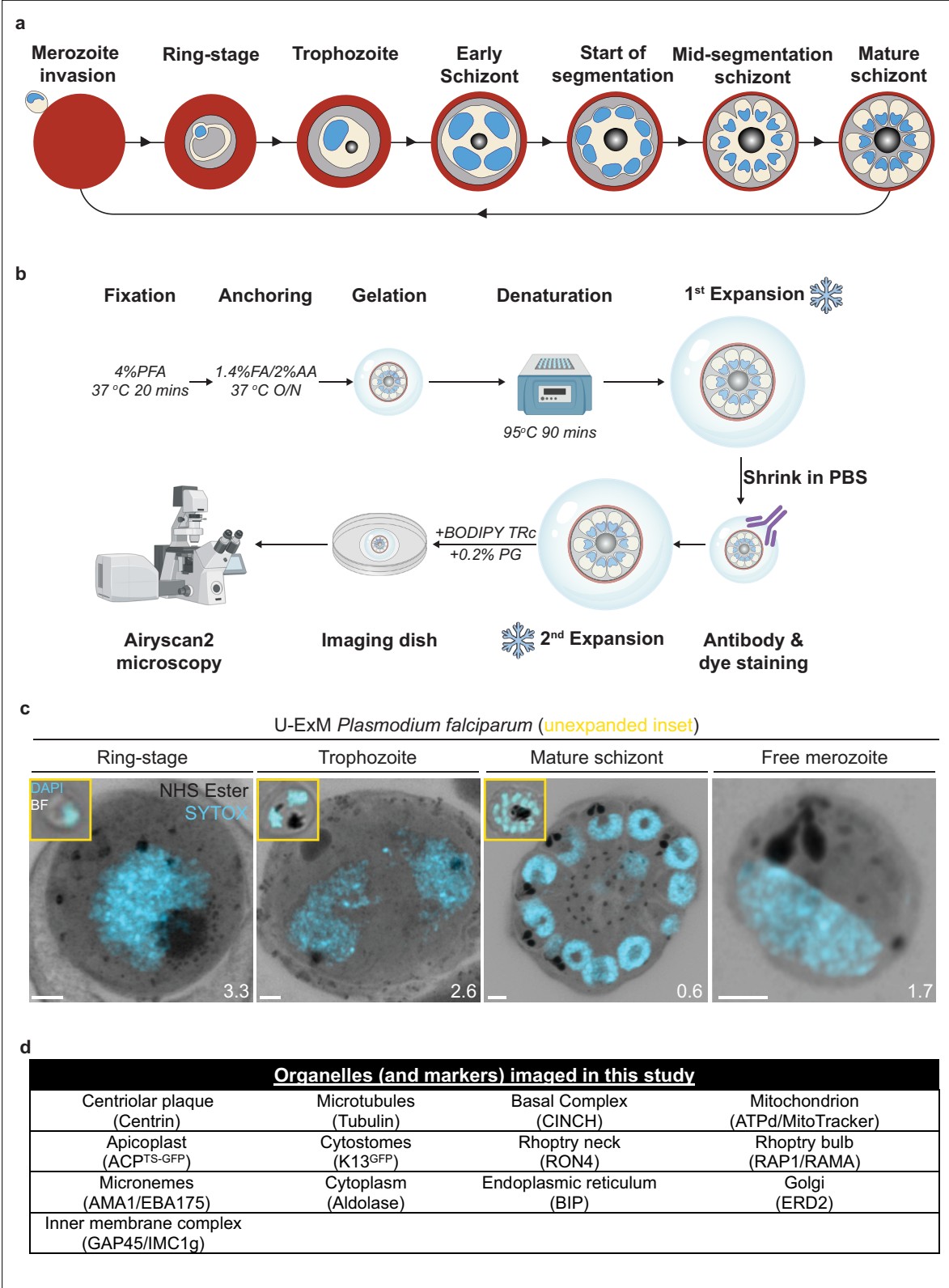

**Figure 1.** Ultrastructural expansion microscopy (U-ExM) workflow and summary of parasite structures imaged in this study. (**a**) Diagram of asexual blood-stage lifecycle of *P. falciparum*. (**b**) U-ExM workflow used in this study. PFA = paraformaldehyde, FA = formaldehyde, AA = acrylamide, PG = propyl gallate. Snowflake indicates steps where gels were cryopreserved. (**c**) Comparison of brightfield and DAPI staining of unexpanded *P. falciparum* parasites (inset) with *P. falciparum* prepared by U-ExM, stained with N-hydroxysuccinimide (NHS) ester (protein density; grayscale) and SYTOX Deep Red

*Figure 1 continued*

(DNA; cyan), and imaged using Airyscan microscopy. Images are maximum-intensity projections, number on image = Z-axis thickness of projection in μm. Scale bars = 2 μm. (**d**) Summary of all organelles, and their corresponding antibodies, imaged by U-ExM in this study.

The online version of this article includes the following figure supplement(s) for figure 1:

**Figure supplement 1.** Size of gels imaged in this study.

**Figure supplement 2.** Cytoplasm staining during intraerythrocytic development.

stages, we set out to determine what parasite structures could, and could not, be visualized by U-ExM when using some of these dyes.

We, and others, have previously shown that combining BODIPY TRc and dye-conjugated NHS ester with U-ExM allows visualization of the parasite plasma membrane (PPM), parasitophorous vacuolar membrane (PVM), nuclear envelope, rhoptries, endoplasmic reticulum (ER), centriolar plaque (CP), basal complex, and apical polar rings (APRs) (*Bertiaux et al., 2021*; *Liffner and Absalon, 2021*; *Simon et al., 2021*). However, many important organelles and parasite structures are either not identifiable using these stains or have yet to be validated, including the mitochondrion, apicoplast, and cytostomes.

To identify and validate the location of as many parasite organelles and structures as possible, we utilized U-ExM coupled with BODIPY TRc, Alexa Fluor 405-conjugated NHS ester (which we will refer to as 'NHS ester'), the nucleic acid (i.e., DNA) stain SYTOX Deep Red, and antibodies directed against 13 different subcellular targets (microtubules, CP, basal complex, inner membrane complex [IMC], mitochondrion, apicoplast, cytostome, rhoptry bulb, rhoptry neck, micronemes, cytoplasm, ER, and Golgi). In this study, all parasites were fixed in 4% paraformaldehyde (PFA), unless otherwise stated, and anchored overnight at 37°C before gelation, denaturation at 95°C, and expansion. Expanded gels were measured (*Figure 1—figure supplement 1*), before shrinking in PBS antibody staining, washing, re-expansion, and imaging (*Figure 1b*). Parasites were harvested at multiple time points during the intraerythrocytic asexual stage and imaged using Airyscan 2 super-resolution microscopy, providing high-resolution three-dimensional imaging data (*Figure 1c*). A full summary of all target-specific stains used in this study can be found in *Figure 1d*.

The first protein we imaged was aldolase, a marker of the parasite cytoplasm (*Figure 1—figure supplement 2*). Aldolase staining was present in all asexual replication stages. During the ring stage, the 'ameboid' shape of the parasite is readily visualized, consistent with previous studies of this stage in time-lapse microscopy of live parasites (*Grüring et al., 2011*; *Figure 1—figure supplement 2*). Regions within the parasite where both aldolase and NHS ester staining were absent are consistent with the expected area of the food vacuole. Typically, the food vacuole would be filled with hemozoin; however, this crystal likely cannot expand and therefore leaves a large space inside the parasite that does not contain significant protein density (*Figure 1—figure supplement 2*).

## The CP, nuclear MTOC, and microtubules

The first major transition during the blood stage of the lifecycle occurs when the parasites turn from rings into trophozoites. Soon after this transition, the parasite will begin to replicate its DNA and undergo mitosis followed by nuclear fission (*Gerald et al., 2011*). Mitosis is coordinated by microtubules, which are in turn nucleated by structures called microtubule organizing centers (MTOCs) (*Sanchez and Feldman, 2017*). *P. falciparum* has a structure known as the CP that spans the nuclear envelope, with intranuclear and cytoplasmic portions (*Simon et al., 2021*). In this study, we will refer to the intranuclear portion of the CP as the inner CP and the cytoplasmic portion of the CP as the outer CP. The inner CP acts as the nuclear MTOC that coordinates *P. falciparum* mitosis, while the function of the outer CP is unknown in asexual blood-stage parasites. The most commonly used MTOC markers are the centrins, which in *Plasmodium* comprise four proteins that localize to the outer CP and appear after intranuclear microtubules have already formed (*Simon et al., 2021*). Given that an MTOC is required for microtubule formation, this implies that an MTOC forms before centrin is visible. To investigate these processes in more detail, we visualized the biogenesis and dynamics of the CP and microtubules during the trophozoite and schizont stages by pairing NHS ester, which we have recently shown can stain both the inner and outer CP (*Liffner and Absalon, 2021*), with an anti-centrin antibody (Clone 20H5, raised against centrin from *Chlamydomonas*) (*Figure 2a*). This antibody

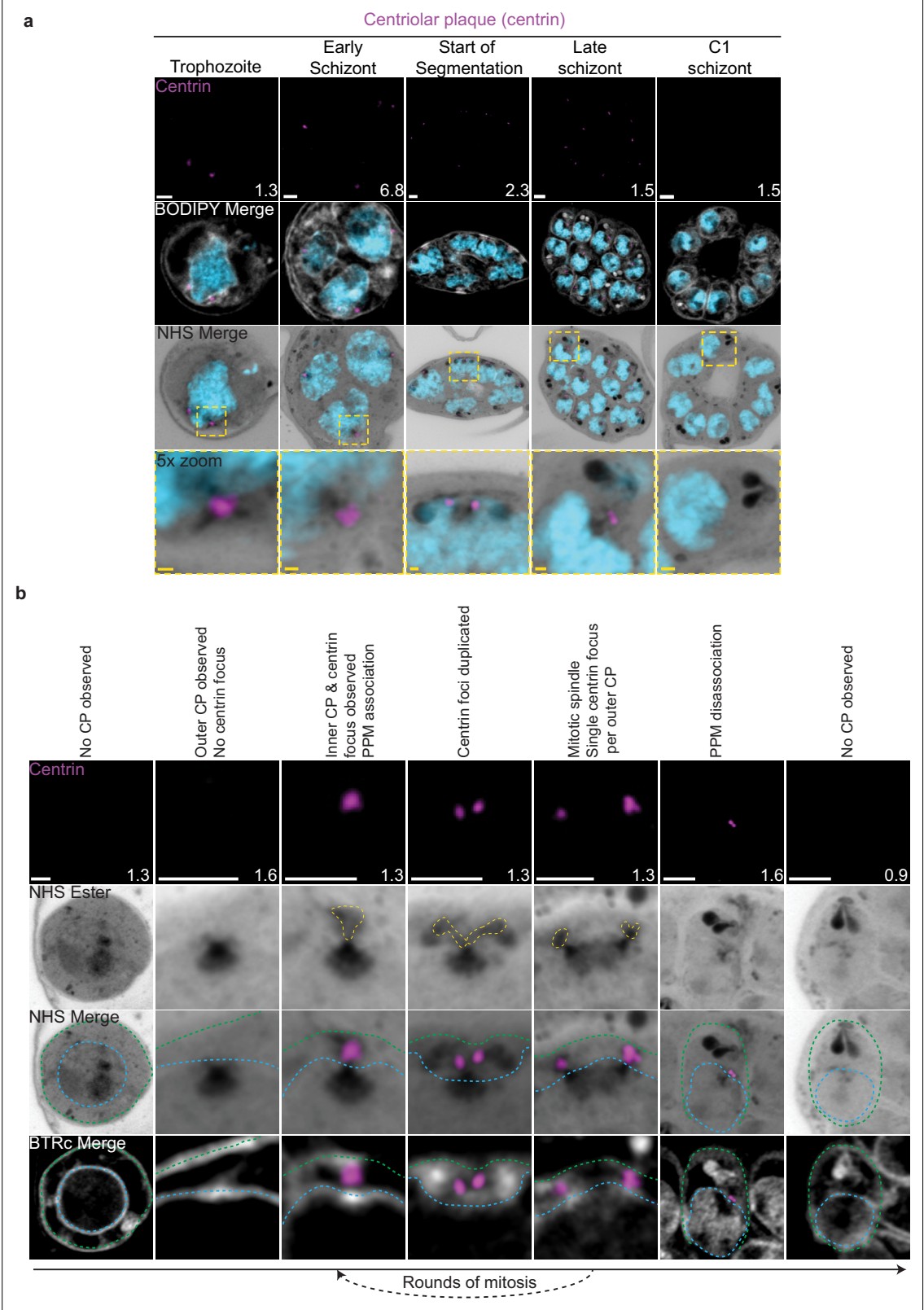

**Figure 2.** Centriolar plaque (CP) biogenesis and dynamics. 3D7 parasites were prepared by ultrastructural expansion microscopy (U-ExM), stained with N-hydroxysuccinimide (NHS) ester (grayscale), BODIPY TRc (white), SYTOX (cyan), and anti-centrin (outer CP; magenta) antibodies and imaged using Airyscan microscopy. (**a**) Images of whole parasites throughout asexual blood-stage development. (**b**) Whole parasite panel (left) followed by individual CP or CP pair zooms following our proposed timeline of events in CP biogenesis, dynamics, and disassembly. Yellow line = cytoplasmic extensions, blue

*Figure 2 continued on next page*

*Figure 2 continued*

line = nuclear envelope, green line = parasite plasma membrane. Images are maximum-intensity projections, number on image = Z-axis thickness of projection in μm. White scale bars = 2 μm, yellow scale bars = 500 nm.

The online version of this article includes the following figure supplement(s) for figure 2:

**Figure supplement 1.** Characterization of outer centriolar plaque (CP) branches.

**Figure supplement 2.** Golgi staining during intraerythrocytic development.

likely recognizes centrin 3 in *P. falciparum* (*Mahajan et al., 2008*), but a recent study suggests that all four *P. falciparum* centrins share an outer CP localization (*Voß et al., 2022*).

## CP biogenesis and disassembly

Neither a recognizable CP nor above-background centrin staining were observed in ring-stage parasites (*Figure 2b*; *Simon et al., 2021*). The inner CP first appeared in mononucleated trophozoites but changed morphology as these parasites got closer to their first nuclear division. In the 23 mononucleated trophozoites we imaged, 52% of CPs lacked cytoplasmic extensions (*Figure 2—figure supplement 1a and b*). These CPs contained only the inner CP and lacked the outer CP as observed by NHS ester (*Figure 2a and b*). All 12 of the trophozoites with an inner CP but no outer CP also lacked centrin staining (*Figure 2—figure supplement 1c*). This matches previous reports that centrin is specifically associated with the outer CP (*Simon et al., 2021*). As expected, this early CP lacking the outer CP was capable of nucleating microtubules (*Figure 2—figure supplement 1a*). The centrin focus and outer CP became visible in mononucleated trophozoites after nucleation of the intranuclear microtubules but prior to the first CP duplication event of the first mitosis (*Figure 2—figure supplement 1a*), consistent with previous reports (*Simon et al., 2021*). The cytoplasmic extensions that form the outer CP began at the nuclear membrane and ended at an NHS ester-dense focus located at the PPM (*Figure 2b*). This association between the outer CP and the PPM has previously been observed in gametocytes and asexual blood stages (*Li et al., 2022*; *Rashpa et al., 2023*), but the temporal nature of this association during the asexual blood stages remained uncharacterized. Our observation of the outer CP using NHS ester, discussed further below, combined with the temporal pattern of outer CP–PPM association, suggests that nuclei are physically anchored to the PPM while parasites are undergoing mitosis.

For as long as parasites continue to undergo mitosis throughout schizogony, the outer CP remains in contact with the PPM (*Figure 2a*). The outer CP appears as one or two elongated bundles, referenced throughout this paper as 'branches,' that stain densely with NHS ester. These branches showed one or two centrin foci that largely matched branch number. That is, of the 139 single-branched outer CPs (outer CPs with a single cytoplasmic extension branch) observed, all had a single centrin focus, and of the 30 double-branched outer CPs observed, 92% had two centrin foci (*Figure 2b*, *Figure 2—figure supplement 1c*). The overwhelming coordination between number of cytoplasmic extension branches and number of centrin foci suggests centrin duplication and duplication of the branches of the outer CP happens quickly and simultaneously. The few cases when there is a mismatch between the number of cytoplasmic extensions and centrin foci all occur in double-branched outer CPs and can be attributed to limitations in our ability to resolve two centrin foci (*Figure 2—figure supplement 1c and d*).

The relative abundance of nuclei with a single CP, defined as a CP not forming a mitotic spindle, versus mitotic CPs, CPs anchored to a mitotic spindle, varied throughout schizogony as did their branch numbers (*Figure 2—figure supplement 1*). As we previously observed that the inner CP forms first, we hypothesized that CPs start out without an outer CP and then develop a single cytoplasmic extension and centrin focus, which duplicate ahead of CP duplication and then segregate with each CP during miotic spindle formation and karyokinesis (*Figure 2b*). This means that during the rapid mitotic events of schizogony that take place at the 6–12 nuclei stage, single outer CPs with a single cytoplasmic extension are very rare. We observed them in just 6% of 221 imaged CPs (*Figure 2—figure supplement 1b*). Single outer CPs with a single cytoplasmic extension were only abundant when the pace of nuclear replication was slow at the early schizont stage (24% of 76 CPs imaged in cells with 2–5 nuclei) and the end of segmentation (44% of 290 CPs imaged in segmenting parasites).

This observed pattern of duplication and segregation also suggests that double-branched outer CPs with two centrin foci have committed to undergoing a new round of mitosis and that the duplication of cytoplasmic extensions represents the first identifiable step in CP duplication. In line with this hypothesis, the most common CP states in schizonts prior to segmentation are single CPs with two cytoplasmic extensions (26% of CPs in cells with 2–5 nuclei, 52% in cells with 6–12 nuclei) that have committed to the next round of mitosis and mitotic CPs with one extension each (42% of CPs in cells with 2–5 nuclei, 36% in cells with 6–12 nuclei) that are finishing a round of mitosis. Interestingly, 8% of CPs observed in the 2–5 nuclei stage and 6% in the 6–12 nuclei stage were mitotic and double-branched, suggesting the duplication of cytoplasmic extensions, and commitment to the next round of mitosis, can happen before karyokinesis is completed (*Figure 1—figure supplements 1e and 2*).

Outer CP branch number reaches semi-synchrony at the beginning of segmentation as defined by the first appearance of a basal complex by NHS ester. At this point, rather than seeing a variety of CP states and branch numbers, virtually all CPs in the same cell share the same mitotic state and branch number. Most CPs are mitotic with a single cytoplasmic extension during early segmentation and then appear as single CP with a single extension during mid-segmentation (*Figure 2—figure supplement 1b*). By the time segmentation is completed, the CP is no longer visible by NHS ester, suggesting that it may disassemble after all mitotic events are finished (*Figure 2b*, *Figure 2—figure supplement 1b*). To ensure imaged parasites were fully segmented, we arrested parasite development by adding the reversible protein kinase G inhibitor compound 1 (C1) (*Collins et al., 2013*; *Gurnett et al., 2002*; *Taylor et al., 2010*). This inhibitor arrests parasite maturation after the completion of segmentation but before egress. When C1 is washed out, parasites egress and invade normally, ensuring that observations made in C1-arrested parasites are physiologically relevant and not a developmental artifact due to arrest. Of 159 nuclei imaged in 6 C1-arrested schizonts, none showed the presence of a CP.

## The apical polar rings, Golgi, and rhoptries are all segregated with the CP

Given the cytoplasmic coordination of mitotic events and the physical tethering of the nucleus to the PPM throughout schizogony, we investigated whether we could observe any coordination extending to the apical organelles and other structures known to be present near the CP at these stages. We observed close association between the outer CP and the rhoptries, Golgi, basal complex, and an apical density reminiscent of the APR.

The number of basal complex structures closely matched the number of outer CP extensions throughout schizogony (156 observed cytoplasmic extensions, 153 observed basal complexes). The basal complex, discussed in detail below, appears as a ring-like structure from early schizogony to the completion of segmentation. No basal complex structures were observed in CPs without cytoplasmic extensions. Of 82 single-branched outer CPs imaged, 81 (99%) showed a single basal complex structure. Of 37 double-branched outer CPs imaged, 33 (89%) showed two basal complex structures. As we observed when imaging centrin foci, the few cases when there is a mismatch between the number of cytoplasmic extensions and basal complex structures mostly occur in double-branched outer CPs. However, in this case, the mismatch cannot be attributed to resolution and most likely reflects a transition state in basal complex replication. CINCH stains the basal complex in a punctate pattern until early segmentation, allowing us to visualize basal complex division events as gaps in the punctate staining (Figure 4a and b). The occasional presence of two cytoplasmic extensions in the absence of these gaps suggests that basal complex division, when visualized by breaks in CINCH staining, is less simultaneous with outer CP branch duplication than centrin focus duplication.

Rhoptry biogenesis, discussed further below, was also closely tied to the number and position of the outer CP. This association of one rhoptry per branch, however, is broken at the start of segmentation, when 205 of 211 imaged CPs (97%) had a rhoptry pair per outer CP branch. Further suggestive of CP–rhoptry interaction is the fact that the rhoptries were positioned immediately next to the outer CP for as long as these were visible by NHS ester. While we had no APR protein marker, the cytoplasmic extensions always ended in an NHS ester-dense focus at the plasma membrane. At the beginning of segmentation, this focus obtained a morphology suggestive of the APR based on its ring shape, position, and size. As described above, a small percentage of parasites commit to the next round of mitosis before they finish segregating their genetic material. In these parasites, CP-associated organelles continued to match the number of cytoplasmic extensions. This gave rise to nuclei that

were anchored to four basal complexes, four single rhoptries, and four proto-APR densities in close proximity (*Figure 4—figure supplement 1*). This matches previous observations by three-dimensional electron microscopy that a single nucleus can have four sets of apical buds (*Rudlaff et al., 2020*).

We also characterized the distribution of the Golgi and its location near the CP and apical organelles, this spatial correlation had been previously described by electron microscopy but has not been thoroughly explored (*Bannister et al., 2000*). The Golgi was visualized using an antibody to ER lumen protein retaining receptor 2 (ERD2), a *cis*-Golgi marker expressed throughout intraerythrocytic development (*Elmendorf and Haldar, 1993*; *Figure 2—figure supplement 2*). Small regions of Golgi were visible at all development stages. In ring-stage parasites and mononucleated trophozoites, one or two Golgi foci were observed near the nucleus but had no clear proximity to the CP. In all five imaged parasites with less than two nuclei and an outer CP, no Golgi was observed near the inner CP. In parasites that had undergone the first round of mitosis and had an outer CP, the Golgi was proximal to the extensions of the outer CP (*Figure 2—figure supplements 1b and 2*) and closely matched their number and presence. Specifically, 21 of 22 imaged parasites with visible outer CPs had Golgi staining near each branch of their outer CP (*Figure 2—figure supplement 1b*). While increased Golgi–CP proximity coincides with the appearance of the cytoplasmic extensions, the Golgi is able to remain toward the apical end of the parasite after these tethers and the CP are no longer visible by NHS ester. However, at this point the Golgi loses its close proximity to the apical organelles and is situated closer to the nucleus. In C1-arrested schizonts, each merozoite has a single Golgi that remains at the apical end of the parasite, typically between the rhoptry bulb and the nucleus (*Figure 2—figure supplement 2*).

In contrast, we did not observe a CP association in the distribution of the ER (*Figure 4—figure supplement 2*) within the parasite. The ER was visualized using an antibody to binding immunoglobulin protein (BIP), a constitutively expressed ER lumen protein (*Kumar et al., 1991*). As expected, ER was detected at all stages of intraerythrocytic development (*Figure 4—figure supplement 2*). In ring-stage and mononucleated trophozoite-stage parasites, the ER could be seen wrapping around the nucleus and forming recognizable cisternae. In multinucleated parasites, the ER was too dense to observe cisternae, but large regions of the cell were occupied by the ER. Following segmentation in C1-arrested schizonts, the ER was only observed contiguous with the nuclear envelope.

Combining the observations that the CP is physically tethered to the PPM through the outer CP and that this anchoring is closely associated with organelles that will define the apical end of the parasite (Golgi, rhoptries, basal complex, and APRs), we suggest that this tethering by the outer CP establishes apical–basal polarity in the parasite early in schizogony. Considering that rhoptries are formed from Golgi-derived cargo (*Ben Chaabene et al., 2021*; *Counihan et al., 2013*), it is unsurprising to find the Golgi forms part of this apical cluster of organelles throughout schizogony. The confined space between nuclear envelope and PPM that these organelles are packed into, for example, may allow each nucleus to provide rhoptry cargo locally to their own rhoptries rather than to all rhoptries in the cell. The same principle could apply to other apical Golgi-derived organelles. However, it remains unclear what role, if any, the outer CP plays in this association, whether any organelles besides the nucleus are physically tethered by these extensions, and how these clusters of organelles remain together during the rapid mitotic events constantly separating sister CPs.

## Characterization of intranuclear microtubules

*P. falciparum* asexual blood stages are known to have two classes of microtubules; intranuclear microtubules, which partake in mitosis (*Liffner and Absalon, 2021*; *Simon et al., 2021*), and subpellicular microtubules (SPMTs), which are cytosolic and extend in a single spine from the apical end of merozoites (*Liffner and Absalon, 2021*; *Simon et al., 2021*). Investigating microtubules with an anti-α-tubulin antibody, we failed to detect microtubules in ring-stage parasites, consistent with previous observations (*Figure 3—figure supplement 1*; *Simon et al., 2021*). Intranuclear microtubules were first visible in mononucleated trophozoite-stage parasites and were present until early segmentation stages, with no intranuclear microtubules visible by the end of segmentation (*Figure 3a*). Intranuclear microtubules arrange into three distinct spindle structures: hemispindles, mitotic spindles, and interpolar spindles. Hemispindles are microtubule structures coming from a single CP that retract prior to CP duplication. Mitotic spindles appear following CP duplication and separate sister chromatids during mitosis. When the two CPs migrate away from each other, they remain connected by an

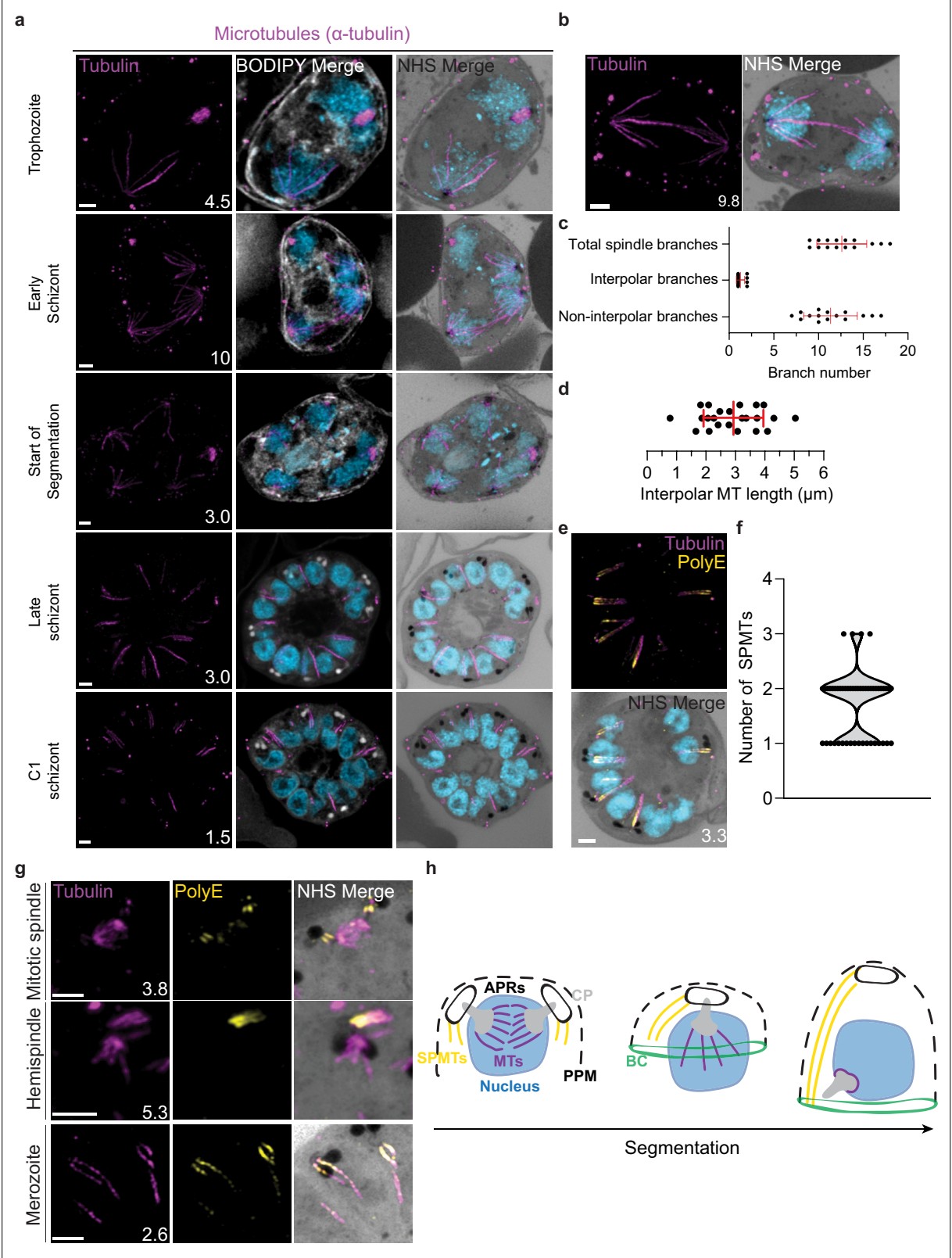

**Figure 3.** Characterization of intranuclear and subpellicular microtubules (SPMTs). 3D7 parasites were prepared by ultrastructural expansion microscopy (U-ExM), stained with N-hydroxysuccinimide (NHS) ester (grayscale), BODIPY TRc (white), SYTOX (cyan), and anti-tubulin (microtubules; magenta) antibodies, and imaged using Airyscan microscopy. (**a**) Images of whole parasites throughout asexual blood-stage development. (**b**) Nuclei in the process of dividing, with their CPs connected by an interpolar spindle. (**c**) The number and type of microtubule branches in interpolar spindles and (**d**)

*Figure 3 continued on next page*

*Figure 3 continued*

length of interpolar microtubules. (**e**) SPMTs stained with an anti-poly-glutamylation (PolyE; yellow) antibody. (**f**) Quantification of the number of SPMTs per merozoite from C1-treated schizonts. (**g**) SPMT biogenesis throughout segmentation. (**h**) Model for SPMT biogenesis. PPM = parasite plasma membrane, APRs = apical polar rings, BC = basal complex, CP = centriolar plaque. Images are maximum-intensity projections, number on image = Z-axis thickness of projection in µm. Scale bars = 2 µm.

The online version of this article includes the following figure supplement(s) for figure 3:

**Figure supplement 1.** Staining of microtubules, centriolar plaque, mitochondrion, and apicoplast in rings or early trophozoite stages.

elongated microtubule structure called the interpolar spindle (or elongated spindle), which retracts prior to nuclear fission (*Liffner and Absalon, 2022*; *Machado et al., 2022*; *Simon et al., 2021*). It has recently been shown that the interpolar spindle is short-lived relative to the hemispindle and mitotic spindle (*Machado et al., 2022*; *Simon et al., 2021*). In this study, we observed 24 interpolar spindles, which allowed us to perform the first detailed characterization of this spindle type (*Figure 3b*).

Interpolar spindles have microtubule branches that connect the two distant CPs (interpolar microtubules), and microtubule branches that do not connect the CPs (non-interpolar microtubules). Each interpolar spindle contained an average of 12.5 (±2.6 SD) total microtubules, of which 1.3 (±0.6 SD) were interpolar microtubules and 11.2 (±2.8 SD) were non-interpolar microtubules (*Figure 3c*). The average number of non-interpolar branches per inner CP was 5.6, which is similar to the previously reported average number of branches in a hemispindle of 5–6 (*Liffner and Absalon, 2021*; *Simon et al., 2021*). This suggests that only the interpolar microtubules retract during the interpolar spindle to hemispindle transition. We measured interpolar microtubules in 3D, adjusting for expansion factor by dividing the measured distance by 4.25, the median expansion factor observed in this study (*Figure 1—figure supplement 1*; 'Materials and methods'). All of the following measurements in this study are reported in this expansion-corrected format. Interpolar microtubules ranged from ~1 to 5 µm, with a mean length of 2.9 µm (±1.0 µm SD) or 12.47 µm before expansion factor correction (*Figure 3d*). In all cases, the CPs connected by interpolar spindles were anchored to the plasma membrane by their cytoplasmic extensions. The large variability in interpolar microtubule size and the continued tethering of the outer CPs to the PPM suggest that interpolar microtubules push PPM-anchored CPs to opposite sides of the cell without causing detachment from the PPM. It is unclear how parasites achieve this sliding effect or how CP-associated organelles are able to retain this association while CPs are moved large distances.

## SPMT length and biogenesis

SPMTs are nucleated in the cytoplasm and have long been observed in merozoites (*Aikawa, 1967*). SPMTs have been shown to be stabilized by polyglutamylation (*Bertiaux et al., 2021*) and can be identified specifically using a combination of anti-tubulin and anti-PolyE antibodies (*Figure 3e*). Using this approach, we characterized 86 SPMTs in 50 merozoites from C1-arrested schizonts. These nascent merozoites had between 1 and 3 SPMTs, with an average of 1.7 (±0.6 SD) (*Figure 3f*). Of 50 imaged merozoites, 48 had at least one SPMT that extended >50% of cell length from the APR to the basal complex. This longest microtubule in a merozoite had an average length of 1.01 µm (±0.24 µm SD). In merozoites with more than one SPMT, the second and third microtubules were shorter than the first, having an average length of 0.8 µm (±0.21 µm SD). Given the large variation in SPMT size and observation that, in segmenting schizonts, the basal end of the SPMTs was in contact with the basal complex throughout segmentation, we hypothesize that most SPMTs measured in our C1-treated schizonts had partially depolymerized. *P. falciparum* microtubules are known to rapidly depolymerize during fixation (*Liffner and Absalon, 2022*; *Simon et al., 2021*). It is unclear, however, why this depolymerization was observed most often in C1-arrested parasites. Thus, we cannot determine whether these shorter microtubules are a by-product of drug-induced arrest or a biologically relevant native state that occurs at the end of segmentation.

Little is known about SPMT biogenesis during the asexual blood stage of *P. falciparum*, but it is currently hypothesized that they are nucleated by the APRs (*Hanssen et al., 2013*; *Morrissette and Sibley, 2002*), as is the case in *Toxoplasma* (*Morrissette and Sibley, 2002*; *Tran et al., 2010*). Curiously, TgCentrin 2 localizes to the APR of *Toxoplasma* tachyzoites (*Hu, 2008*), but no Centrin 2 has been observed to localize to the APRs of *P. falciparum*. Furthermore, it was recently shown that the SPMTs of *P. falciparum* gametocytes, which lack an APR, are formed at the outer CP, in the space

between the nuclear envelope and PPM (*Li et al., 2022*). Leveraging our ability to specifically detect SPMTs using PolyE, we investigated the possibility that merozoite SPMTs are also formed at the outer CP and subsequently transferred onto the APR during segmentation. In schizonts where nuclei are approaching or have completed their final mitosis (~15n), we observed small cytoplasmic microtubules that stained strongly with PolyE appear in the area between the outer CP and PPM (*Figure 3g and h*). However, we did not achieve a resolution that allowed us to distinguish individual APRs or to confidently pinpoint whether the microtubules were nucleated at the APRs or the cytoplasmic extensions. Likely, higher resolution imaging techniques are needed to resolve the site of SPMT nucleation in merozoites.

## Segmentation machinery (IMC and basal complex)

Following replication of their genetic material during the trophozoite and early schizont stages, parasites partition their nuclei and organelles into ~30 daughter merozoites from the common cytoplasm of a schizont (*Francia and Striepen, 2014*). This form of cytokinesis, called segmentation, takes place in the final hours of schizogony and culminates with the physical separation of each daughter cell and their egress from the host RBC. The IMC is a double lipid bilayer formed from flattened vesicles that scaffolds the process of segmentation as well as anchors many proteins important for parasite shape and motility (*Harding and Meissner, 2014*).

## The IMC cannot be distinguished from the plasma membrane by U-ExM

The IMC forms de novo during segmentation starting at the apical end of the parasite, where the outer CP is anchored to the plasma membrane (*Figure 4—figure supplement 3a*; *Harding and Meissner, 2014*). This can be observed using the IMC-anchored protein glideosome-associated protein 45 (GAP45), which bridges the IMC and plasma membrane, as well as using BODIPY TRc, which shows increased membrane staining in the area overlapping GAP45 (*Jones et al., 2006*; *Figure 4—figure supplement 3a*). As segmentation progresses, the IMC expands around the nucleus and associated organelles until it envelops the daughter cell, leaving an opening at the apical end, where the APR is located, and the basal end, where the basal complex resides (*Figure 4—figure supplement 3a*). While the pellicle was easily visualized as a whole, we were unable to distinguish the IMC membranes from the PPM (*Figure 4—figure supplement 3b and c*). We stained parasites using the plasma membrane marker MSP1 and two different IMC markers: GAP45, which lies between the IMC and PPM, and IMC1g, which is attached to the cytoplasmic face of the IMC (*Blackman et al., 1994*; *Cepeda Diaz et al., 2023*; *Kono et al., 2012*). In both cases, we were unable to resolve the IMC marker from MSP1 (*Figure 4—figure supplement 3b and c*).

## Basal complex dynamics throughout segmentation

The basal complex is an essential ring structure located at the basal end of the IMC (*Morano and Dvorin, 2021*). It is hypothesized to act as a contractile ring that guides IMC biogenesis and mediates abscission of newly formed merozoites by separating the IMC and plasma membrane from the residual body. We used parasites where PfCINCH, a basal complex marker, was tagged with a spaghetti monster V5 (smV5) tag to follow basal complex development throughout schizogony (*Figure 4a*; *Rudlaff et al., 2019*). CINCH is first visible at early schizogony (3–5 nuclei stage) as a small ring-like structure surrounding an NHS ester-dense focus on the plasma membrane that is tethered to the CP (*Figure 4b*). Of 55 early schizont CPs imaged, 44 (80%) had matching numbers of outer CP branches and basal complex structures. This suggests that as CPs divide, they each inherit a CINCH ring that has been split by the duplication of the cytoplasmic tethers (*Figure 4b*). Early IMC proteins have been described to form cramp-like structures like these prior to attaining their characteristic ring structure later in schizogony (*Hu et al., 2010*; *Kono et al., 2012*). During the rapid nuclear divisions of schizogony, 77% of the CINCH structures of mitotic CPs show a break in the ring (*Figure 4—figure supplement 1b*). This break faces a sister basal complex with its own cytoplasmic extension (*Figure 4b*, *Figure 4—figure supplement 1b*). These observations suggest that, upon duplication of the outer CP branches, the basal complex ring likely 'breaks' into two semicircles, which re-seal to form their own ring prior to the next branch duplication (*Figure 4b*, *Figure 4—figure supplement 1b*).

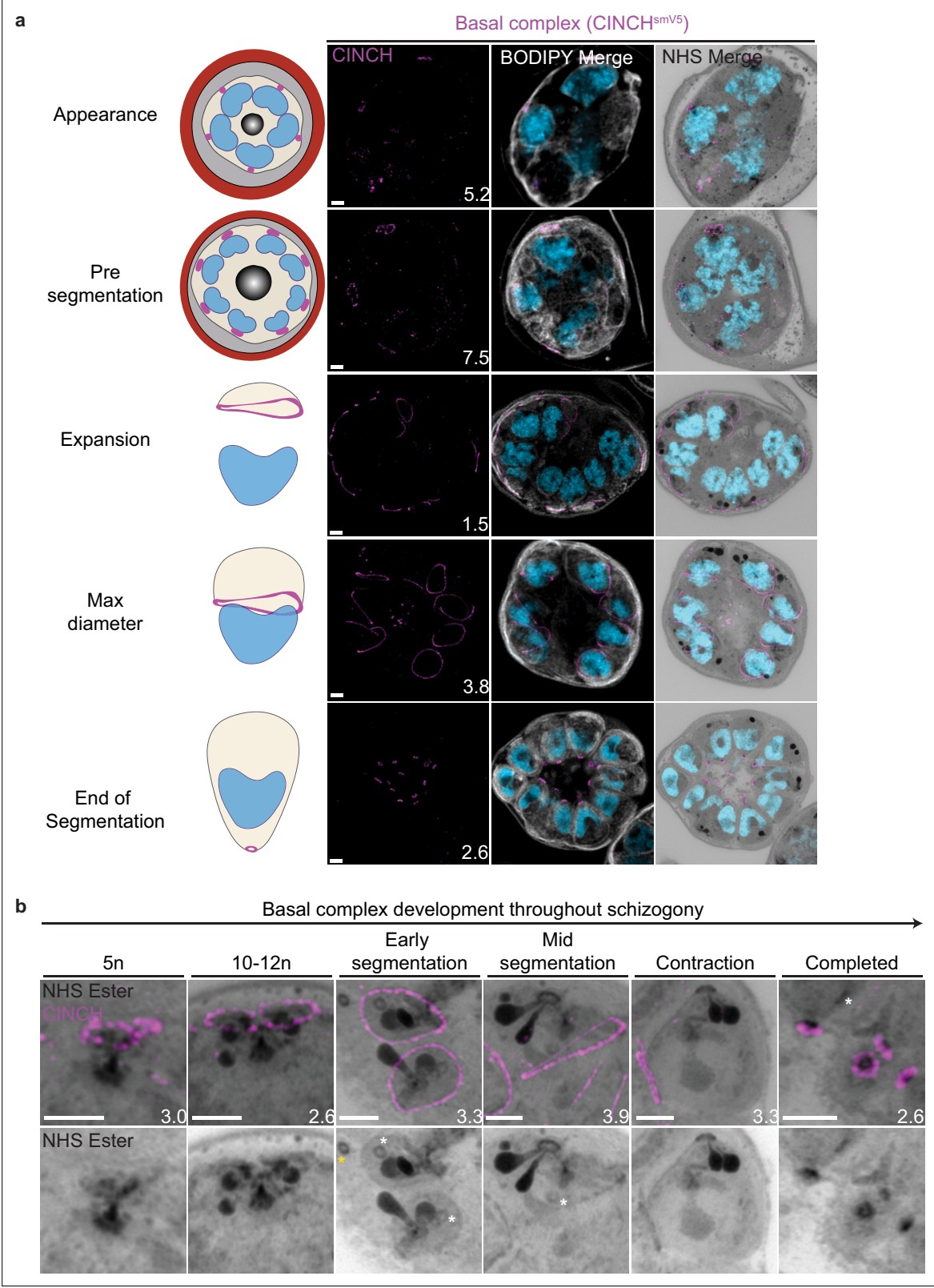

**Figure 4.** Basal complex biogenesis and development throughout segmentation. Parasites expressing an smV5-tagged copy of the basal complex marker CINCH were prepared by ultrastructural expansion microscopy (U-ExM), stained with N-hydroxysuccinimide (NHS) ester (grayscale), BODIPY TRc (white), SYTOX (cyan), and anti-V5 (basal complex; magenta) antibodies and imaged using Airyscan microscopy across segmentation. (**a**) Images of whole parasites throughout asexual blood-stage development. (**b**) Basal complex development during schizogony. The basal complex is formed around

*Figure 4 continued on next page*

*Figure 4 continued*

the parasite plasma membrane (PPM) anchor of the outer centriolar plaque (CP). In nuclei whose outer CP has two branches, and will therefore undergo mitosis, the basal complex rings are duplicated. From early segmentation, the basal complex acquires a stable, expanding ring form. Cytostomes that will form part of merozoites are marked with a white asterisk, while those outside merozoites are marked with a yellow asterisk. Images are maximum-intensity projections, number on image = Z-axis thickness of projection in µm. Scale bars = 2 µm.

The online version of this article includes the following figure supplement(s) for figure 4:

**Figure supplement 1.** Basal complex biogenesis in mitotic nuclei.

**Figure supplement 2.** Endoplasmic reticulum staining during intraerythrocytic development.

**Figure supplement 3.** Inner membrane complex (IMC) progression through segmentation.

Once segmentation begins and the outer CP stops duplicating, CINCH forms a bona fide ring with smooth borders (*Figure 4*). This matches previously reported behavior of early IMC proteins and supports our hypothesis that the cramp-like structures arise from IMC and basal complex division. At this point, nuclei reach a point of semi-synchronicity. All 64 imaged CPs in early segmentation parasites had duplicated and were forming a mitotic spindle. This event marks the last nuclear division the parasite will undergo. Each of these CPs had a single uninterrupted basal complex ring (*Figure 4b*). Thus, we cease to observe events where a single nucleus is attached to four basal complexes as CPs have ceased committing to future rounds of mitosis. As the parasite undergoes segmentation, the basal complex expands and starts moving in the basal direction. By the time the basal complex reaches its maximum diameter, all nuclear divisions have been completed, each nucleus has a single CP and basal complex, and no mitotic spindles are visible (*Figure 4*). After this point, the basal complex contracts and continues to move away from the apical end. By the time segmentation is completed, the basal complex is an NHS ester-dense ring that is smaller than the APR.

## NHS ester as a basal complex marker

While the basal complex stains brightly with NHS ester at the end of segmentation (*Figure 4*), this staining is not consistent throughout schizogony. NHS ester staining of the basal complex is not visible or is very faint during early schizogony. Once the basal complex attains its bona fide ring form during early segmentation, it stains reliably, though faintly, with NHS ester. This staining intensifies after the basal complex begins to contract. The denser staining observed during basal complex contraction could be due to recruitment of more basal complex proteins at the midpoint of schizogony, an increase in protein density as the ring area decreases during contraction, or both. Once the parasites finish segmentation, the basal complex is at its brightest (*Figure 4b*). While NHS ester staining correlates with CINCH, it does not perfectly overlap with it. CINCH consistently appears as a larger ring with a slight basal shift relative to NHS ester after the basal complex reaches maximum diameter, an effect most visible at the end of segmentation (*Figure 4b*). Since this shift is consistent with parasite anatomy regardless of parasite orientation, it suggests it is not an imaging artifact. There is no primary antibody against CINCH at this point, and so it is not possible to determine whether the lack of overlap with NHS ester is due to distance between the smV5 tag and the main protein density of CINCH (CINCH is 230 kDa). It is also possible that this difference in localization reflects basal complex architecture similar to that previously observed in *Toxoplasma gondii*, where the basal complex consists of multiple concentric rings (*Anderson-White et al., 2012*; *Engelberg et al., 2022*; *Hu, 2008*; *Roumégous et al., 2022*).

## Mitochondrion and apicoplast

The apicoplast and mitochondrion undergo pronounced morphological changes during the *P. falciparum* blood-stage lifecycle (*van Dooren et al., 2005*; *van Dooren et al., 2006*). Both are long, and often branching, organelles whose complex three-dimensional morphologies have only been robustly studied using electron microscopy-based techniques (*Rudlaff et al., 2020*).

## Looped regions of the mitochondrion display low membrane potential

To visualize the mitochondria, we stained live parasites using MitoTracker Orange CMTMRos prior to fixation and expansion (*Figure 5—figure supplement 1a*). MitoTracker Orange CMTMRos accumulates in live mitochondria, driven electrophoretically by membrane potential, and is retained after

fixation (*Elmore et al., 2004*; *Poot et al., 1996*). When imaged at high resolution, MitoTracker can be used to observe individual cristae in the mitochondria of mammalian cells (*Wolf et al., 2019*). *Plasmodium* cristae morphology is different from that found in mammalian mitochondria; cristae are thought to be bulbous or tubular rather than lamellar and are present in gametocytes but absent from asexual blood-stage parasites (*Evers et al., 2021*; *Evers et al., 2023*). To our surprise, rather than showing continuous staining of the mitochondria, MitoTracker staining of our expanded parasites revealed alternating regions of bright and dim staining that formed MitoTracker-enriched pockets (*Figure 5— figure supplement 1b*). These clustered areas of MitoTracker staining were highly heterogeneous in size and pattern. Small staining discontinuities like these are commonly observed in mammalian cells when using MitoTracker dyes due to the heterogeneity of membrane potential from cristae to cristae as well as due to fixation artifacts. At this point, we cannot determine whether the staining we observed represents a true biological phenomenon or an artifact of this sample preparation approach. Our observed MitoTracker-enriched pockets could be an artifact of PFA fixation, a product of local membrane depolarization, a consequence of heterogeneous dye retention, or a product of irregular compartments of high membrane potential within the mitochondrion, to mention a few possibilities. Further research is needed to conclusively pinpoint an explanation.

In addition to these small staining discontinuities, we observed large gaps in MitoTracker staining within parasites at all stages of development. This included pre-segmentation parasites, where we would expect a single continuous mitochondrion to be present (*Figure 5—figure supplement 1a*). To our knowledge, no membrane potential discontinuities or fixation artifacts of this size have been reported in mammalian cells. So, as a secondary way to visualize the mitochondria and better characterize these staining discontinuities, we generated a transgenic cell line with the putative ATP synthase F0 subunit-d (ATPd, Pf3D7_0311800) tagged with a spaghetti monster HA tag (*Viswanathan et al., 2015*; *Figure 5—figure supplement 2*). ATPd is a membrane-embedded proton channel that had not previously been localized to the mitochondria in *P. falciparum* but was identified as a mitochondrial protein in a recent proteomics study (*van Esveld et al., 2021*; *Evers et al., 2021*). Furthermore, its *Toxoplasma* homolog has been shown to localize to the mitochondria (*Barylyuk et al., 2020*; *Sheiner et al., 2011*). We confirmed that ATP synthase subunit F0 localizes to *P. falciparum* mitochondria, as it largely co-localized with MitoTracker staining, forming a border around it due to its membrane association (*Figure 5a*, *Figure 5—figure supplement 1a*). ATPd, like MitoTracker, had a heterogeneous distribution throughout the mitochondria, but it did not show the same large gaps in staining. ATPd allowed us to better visualize regions of the mitochondria that appeared to fold onto themselves and fuse with each other, as has been previously described (*van Dooren et al., 2005*). Thus, MitoTracker and ATPd are both useful but imperfect markers for the mitochondria, with neither of them showing a continuous, even distribution throughout the organelle.

Curiously, 25 of 26 imaged parasites showed MitoTracker discontinuities specifically in regions where the ATPd signal formed looped structures (*Figure 5b*). These structures were defined as areas where the mitochondria showed a turn or fold of ~180°. Of the 41 looped regions identified, 75% lacked MitoTracker staining. This suggests that mitochondria looped regions in *P. falciparum* have some degree of depolarization that prevents MitoTracker accumulation or that MitoTracker initially accumulates in these regions but is not bound and retained. The biological significance of these areas, if any, is currently unclear.

## Growth of the apicoplast and mitochondrion

To visualize the apicoplast, we utilized a previously established cell line that expresses GFP fused to the apicoplast transit peptide of acyl carrier protein (ACP) (*Florentin et al., 2020*), which we will refer to as apicoplast-GFP (*Figure 6a*). This marker allowed for a relatively even and continuous staining of the organelle. We quantified mitochondrion and apicoplast signal area using ATPd-smHA and apicoplast-GFP respectively as a proxy measurement of size (*Figures 5c and 6b*). Tracking this in parallel to parasite nucleus number allowed us to determine whether the growth of these organelles occurred progressively with simultaneous rounds of mitosis and nuclear division. In mononucleated ring and trophozoite parasites, both the mitochondria and the apicoplast are relatively small, having an average area of 13.39 μm² (±15.1 μm² SD) and 4.81 μm² (±2.62 μm² SD), respectively, in expanded parasites (*Figures 5c and 6b*). As expected from live cell observations (*van Dooren et al., 2005*), both organelles show significant growth and spread throughout the cell in multinucleated parasites, adopting an

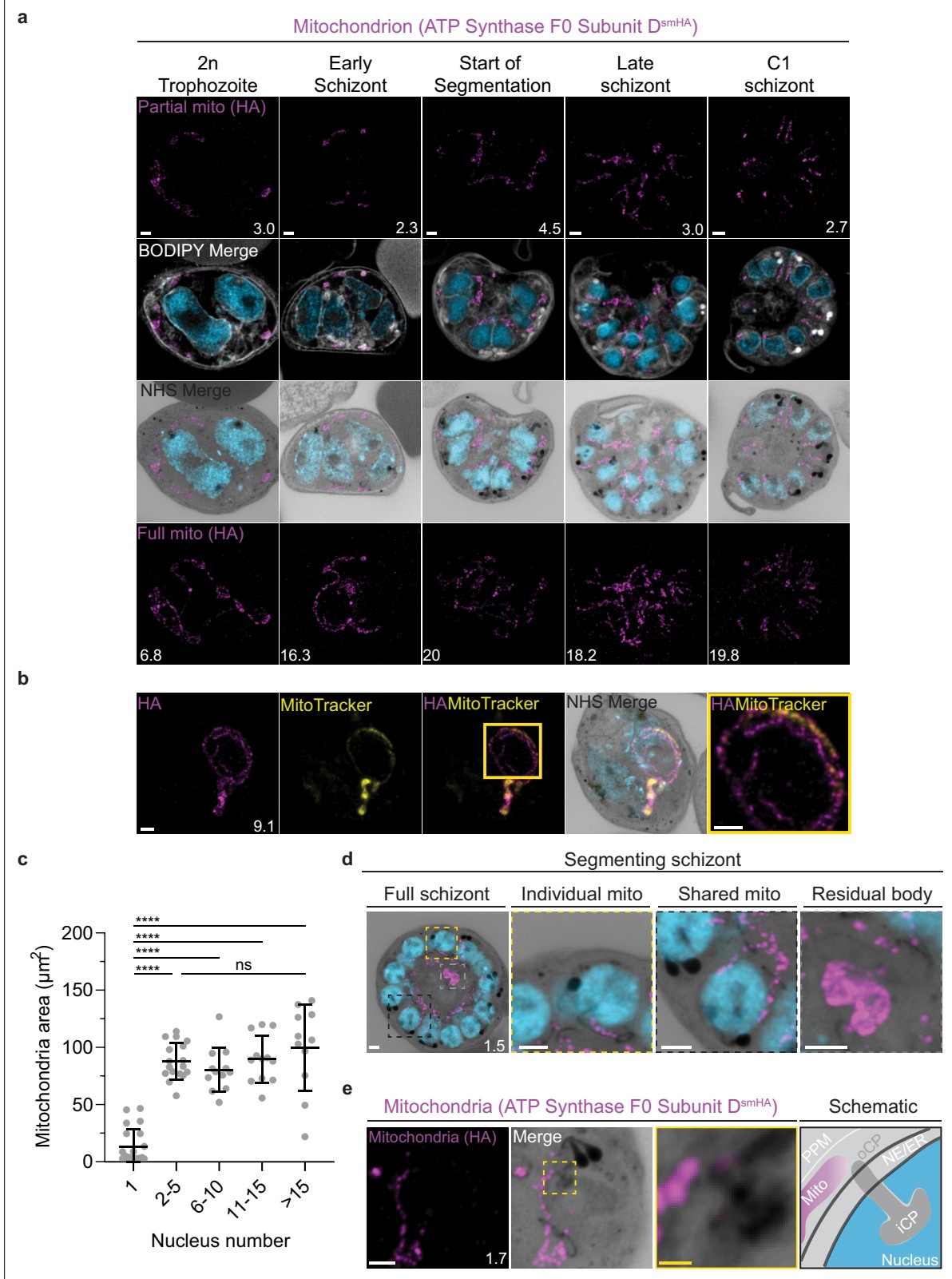

**Figure 5.** Growth and fission of the mitochondrion. Parasites with an smHA-tagged copy of the ATP Synthase F0 Subunit D (ATPd, Pf3D7_0311800) as a mitochondrial marker were prepared by ultrastructural expansion microscopy (U-ExM), stained with N-hydroxysuccinimide (NHS) ester (grayscale), BODIPY TRc (white), SYTOX (cyan), and anti-HA (mitochondrion; magenta) antibodies and imaged using Airyscan microscopy. (**a**) Images of whole parasites throughout asexual blood-stage development. Maximum-intensity projections of both a subsection of the cell (partial mito) and the full cell

*Figure 5 continued on next page*

*Figure 5 continued*

(full mito) are shown. (**b**) ATPd staining was compared against MitoTracker Orange CMTMRos (yellow), which showed discontinuous staining in looped regions. (**c**) Area of the mitochondrion was quantified for parasites of varying nucleus number. Seventy-three cells were counted across four biological replicates. ****p<0.001, ns = p >0.05 by one-way ANOVA, error bars = SD. (**d**) Schizont with mitochondria that have undergone fission (yellow zoom), mitochondria that are shared between two nascent merozoites (black zoom), and mitochondria left outside merozoites in the forming residual body (gray). (**e**) During fission, mitochondria associate with the outer centriolar plaque (oCP). Images are maximum-intensity projections, number on image = Z-axis thickness of projection in μm. White scale bars = 2 μm, yellow scale bars = 500 nm.

The online version of this article includes the following figure supplement(s) for figure 5:

**Figure supplement 1.** Comparison of MitoTracker with ATPd staining and residual body (RB) mitochondria quantification.

**Figure supplement 2.** Generation of ATPd (Pf3D7_0311800) smHA parasites.

elongated and branching morphology (*Figures 5 and 6*). Mitochondria grow almost exclusively during the first two rounds of nuclear replication, achieving an average size of 87.9 μm² (±15.9 μm² SD) at the 2–5 nuclei stage. This size remains relatively constant until segmentation, with the average mitochondria size right before the start of fission being 99.6 μm² (±37.4 μm² SD) (*Figure 5c*). In contrast, the apicoplast continues to grow past the 2–5 nuclei stage, having an average size of 25.5 μm² (±7.21 μm² SD) at the 2–5 nuclei stage and 34.97 μm² (±11.23 μm² SD) in cells with >15 nuclei (*Figure 6b*). These data suggest that the mitochondrion and apicoplast do not grow simultaneously with or as a response to nuclear replication during schizogony. Rather, both organelles show the largest increase in size during the 1–2 nuclei transition and either plateau in size, in the case of the mitochondria, or enter a second phase of slower growth that ends shortly before segmentation, in the case of the apicoplast.

## Fission of the mitochondrion and apicoplast

*P. falciparum* has a single, large, branching, mitochondrion and apicoplast throughout most of the asexual blood stage (*van Dooren et al., 2005*; *van Dooren et al., 2006*; *Verhoef et al., 2021*). During segmentation, however, these organelles undergo fission such that each merozoite inherits an individual apicoplast and mitochondrion (*Rudlaff et al., 2020*; *van Dooren et al., 2005*). While it has been shown that apicoplast fission occurs before mitochondrial fission, it is unclear how fission occurs (*Rudlaff et al., 2020*). A recent review (*Verhoef et al., 2021*) posed three possible mechanisms: synchronous fission where the organelle simultaneously divides into all daughter parasites at once, outside-in fission where fission occurs at the ends of the organelle, or branching point fission where a first fission event divides the organelle into larger segments and a subsequent fission event leaves each merozoite with an individual organelle (*Verhoef et al., 2021*). It also remains unclear how accurate segregation into daughter cells is monitored. In *T. gondii*, the apicoplast associates with the centrosomes prior to undergoing fission. A similar association between apicoplasts and CPs has been proposed in *Plasmodium* but still lacks evidence due to the difficulty of observing the *Plasmodium* CP in live cells (*van Dooren et al., 2005*). The mitochondrion is not thought to associate with the CP in *Toxoplasma* or *Plasmodium,* and its mechanism for ensuring accurate segregation remains unknown.

In the process of imaging the mitochondria and apicoplasts of segmenting parasites, we observed a transient outer CP association prior to and during fission in both organelles. Just before the start of segmentation, there is little association between the outer CP and apicoplast. Of five parasites imaged that had >10 nuclei but had not yet started segmentation, three had no contact points between the outer CP and apicoplast branches and the other two had <4 contact points. Once segmentation starts, all apicoplast branches contact an outer CP each and remain in contact with the outer CP until the end of apicoplast fission and CP degradation (*Figure 6c*). In 11 imaged segmenting schizonts, all outer CPs showed contact with an apicoplast branch each. This association starts in early segmentation, when a single branched apicoplast connects all outer CPs. By the time the basal complex reaches maximum diameter and begins contraction, we observe seven parasites where all branches have completed fission and two parasites that had at least one or more apicoplast segments still connecting multiple merozoites (*Figure 6c*). Mitochondria fission follows a very similar pattern, but later in parasite development. Prior to the basal complex reaching maximum diameter, we observe no significant connection between the outer CP and mitochondria. When the basal complex entered its contraction phase, we observed nine parasites where all outer CPs were in contact with one branch of the mitochondria each (*Figure 5d and e*). Only one of these nine parasites had an intact pre-fission mitochondrion, while the other eight had undergone at least one fission event. Matching previous

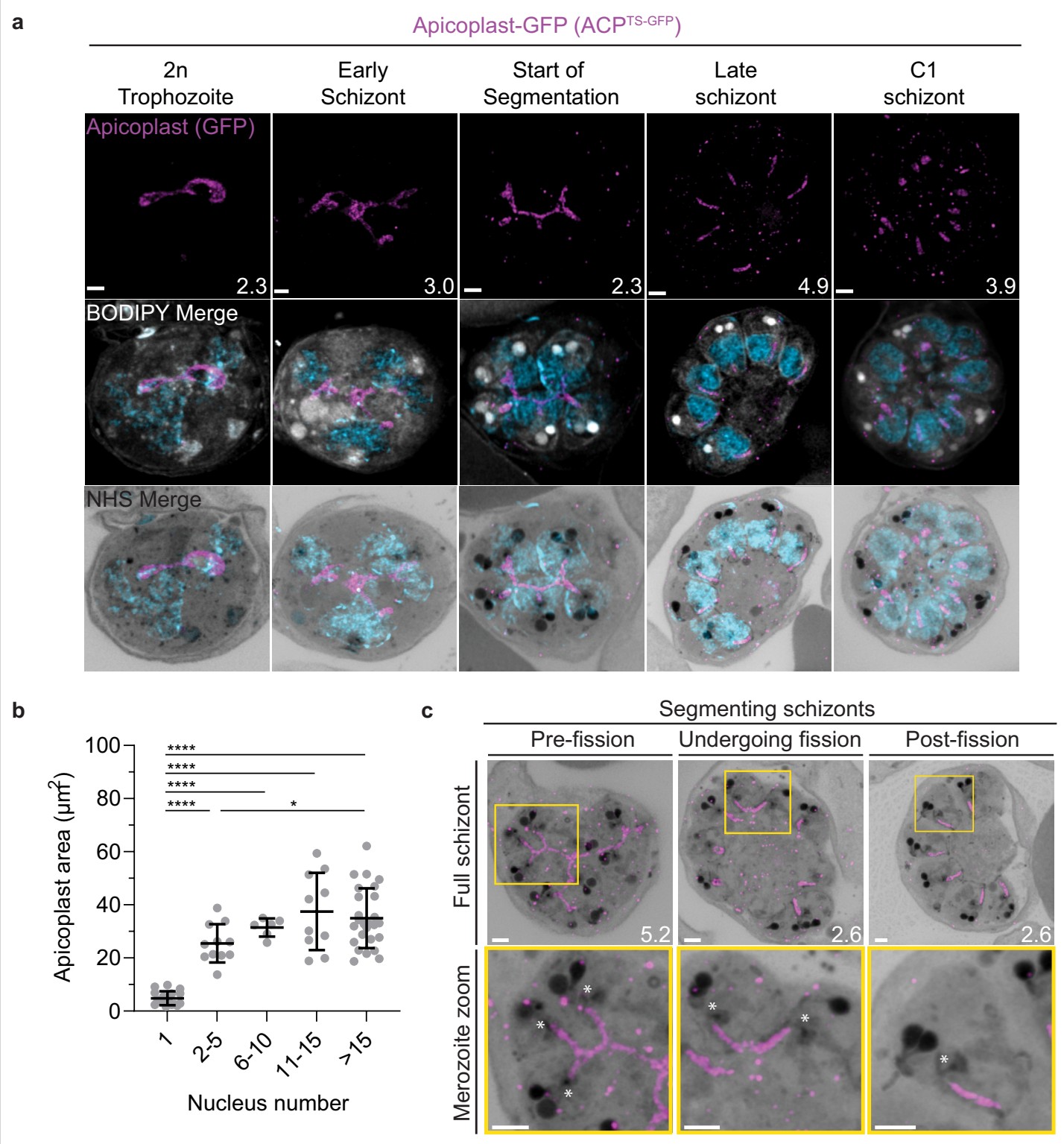

**Figure 6.** Growth and fission of the apicoplast. Parasites expressing GFP-conjugated to the apicoplast transit signal of ACP (ACP^TS-GFP) were prepared by ultrastructural expansion microscopy (U-ExM), stained with N-hydroxysuccinimide (NHS) ester (grayscale), BODIPY TRc (white), SYTOX (cyan), and anti-GFP (apicoplast) (magenta) antibodies and using Airyscan microscopy. (**a**) Images of whole parasites throughout asexual blood-stage development. Maximum-intensity projections of both a subsection of the cell (partial mito) and the full cell (full mito) are shown. (**b**) Area of the apicoplast was quantified for parasites of varying nucleus number. Seventy cells were counted across three biological replicates. ****p<0.001, *p<0.05 by one-way ANOVA, error bars = SD. (**c**) Representative images of the different stages of apicoplast fission. Images are maximum-intensity projections, number on image = Z-axis thickness of projection in μm. Asterisks represent centriolar plaques. Scale bars = 2 μm.

descriptions of mitochondria and apicoplast segmentation, C1-arrested schizonts having completed segmentation show elongated mitochondria (*Figure 5a*) and small, rounded apicoplasts (*Figure 6a*).

Both mitochondria and apicoplast fission showed neighboring nascent merozoites that shared a single branch of mitochondria or apicoplast passing through both of their basal complexes while others had an individual mitochondrion or apicoplast that had already separated from the rest (*Figures 5d and 6c*). This suggests that fission does not occur synchronously (*Figure 5d*) and supports the model of branching point fission. In other words, parasites seem to undergo a primary fission event that leaves only some merozoites sharing stretches of the organelles and then a subsequent fission event leaves each merozoite with an individual apicoplast and mitochondrion (*Figure 5e*). Unfortunately, BODIPY TRc does not distinctly stain the membranes of the mitochondria and apicoplast. So, it is not possible for us to determine whether the observed breaks in staining of our chosen organelle markers truly indicate a complete fission of the mitochondria or apicoplast membranes. Thus, while suggestive of branching point fission, our data is not sufficient to conclusively determine the sequence of fission events in these organelles. More research with additional mitochondrial and apicoplast markers is needed to confirm the observations made in this study and conclusively map out the growth and fission of these organelles.

## Characterization of residual body mitochondria

At the completion of segmentation, the parasite forms a structure known as the residual body, which contains parasite material, such as the hemozoin crystal, that was not incorporated into merozoites during segmentation (*Rudlaff et al., 2020*). The residual body is poorly understood in *Plasmodium*, but in *Toxoplasma* it has been shown that a significant amount of the mitochondria, and not the apicoplast, is left behind in the residual body following segmentation (*Nishi et al., 2008*).

There is no well-characterized marker of the residual body in *Plasmodium*. So, for this study, we defined the residual body as any area within the parasitophorous vacuole membrane but visibly external to any merozoite in a C1-arrested schizont as determined by BODIPY TRc staining. We imaged 35 C1-arrested schizonts and observed that 54% had mitochondrial staining inside the residual body (*Figure 5d*, *Figure 5—figure supplement 1c*). To determine the proportion of total mitochondria that gets included in the residual body, we quantified the fluorescence of both mitochondria in the residual body and mitochondria in merozoites. Of the 19 parasites that showed mitochondria staining inside the residual body, the amount of material ranged from 1 to 12% of the total mitochondrial staining in the parasite (*Figure 5—figure supplement 1d*). On average, the residual body had approximately 1.5× more mitochondrial staining than the average merozoite (*Figure 5—figure supplement 1e*). No significant apicoplast staining was ever observed in the residual body, similarly to what has been reported for *Toxoplasma* (*Nishi et al., 2008*).

## Cytostomes

During its intraerythrocytic development, *P. falciparum* engulfs host cell cytoplasm from which it catabolizes hemoglobin as a source of amino acids (*Francis et al., 1997*). The parasite is separated from its host cell by the parasitophorous vacuole, and therefore the uptake of host-cell cytosol requires invagination of both the PPM and PVM. The cytostome coordinates this endocytic process and is comprised of two key regions: a protein-dense collar region, which forms the pore through which membrane invagination will occur, and the membranous bulb region, which contains the RBC-derived cargo (*Milani et al., 2015*; *Xie et al., 2020*).

## NHS ester staining reveals pore-like structures at the PPM

Prior to this study, cytostomes were not immediately obvious by NHS ester staining given the large number of features that were visible using this stain but pending validation. While observing the basal complex of segmenting schizonts (*Figure 4a*), we noticed that merozoites contained a second NHS ester-dense ring (*Figure 7a*). The size and position of this NHS ester ring matched that of an endocytic micropore recently identified in *Toxoplasma* tachyzoites (*Koreny et al., 2022*). In that study, the micropore was identified using Kelch13 (K13) as a marker (*Koreny et al., 2022*). The *Plasmodium* equivalent to this K13 micropore is the cytostome, so to determine whether this NHS ester-dense ring was indeed a cytostome, we evaluated a parasite strain where the endogenous K13 was fused to GFP

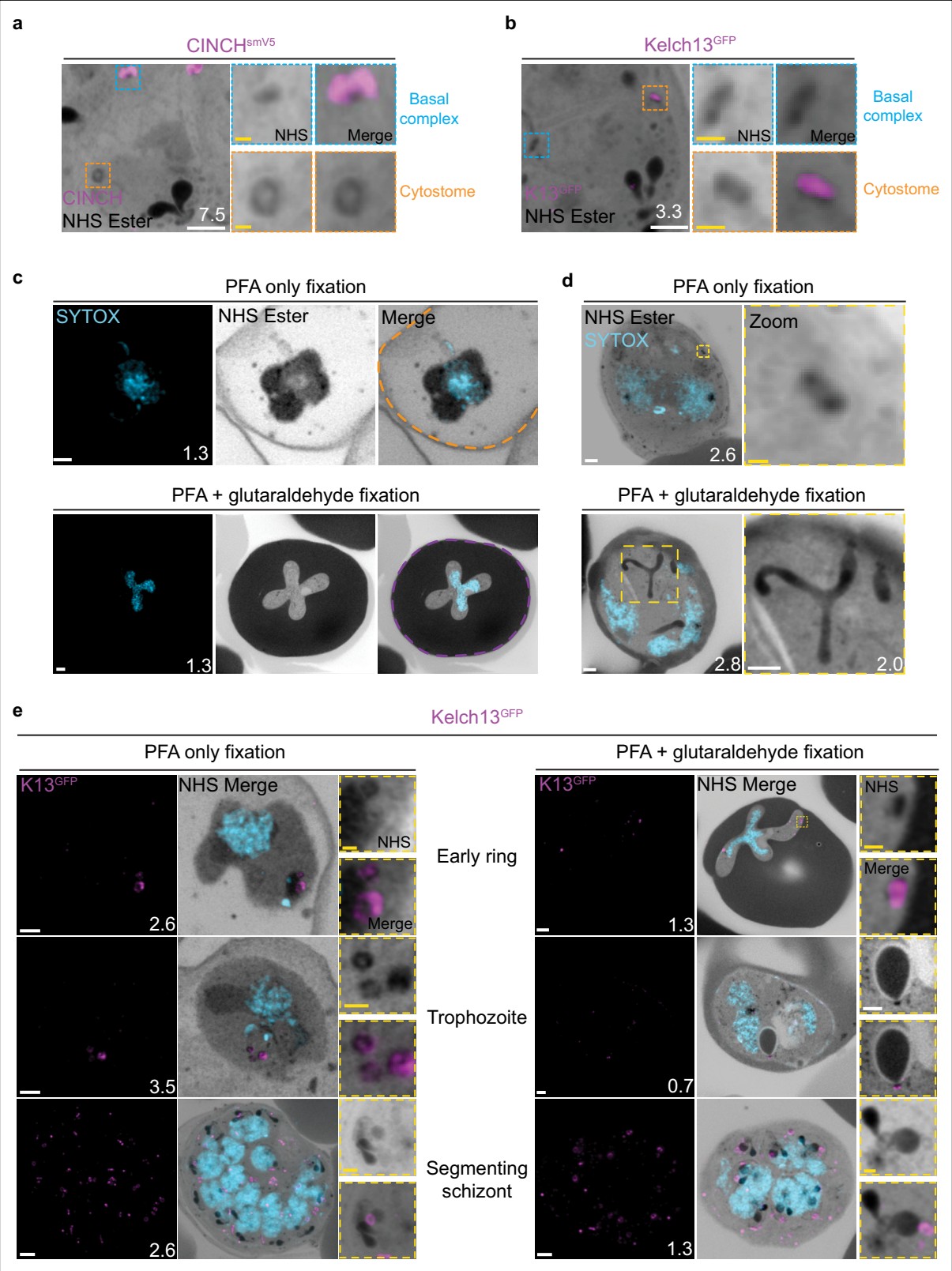

**Figure 7.** Cytostomes are observable by ultrastructural expansion microscopy (U-ExM) throughout the asexual blood stage of the lifecycle. (**a**) Parasites expressing an smV5-tagged copy of the basal complex marker CINCH, prepared by U-ExM, and stained with anti-V5, show N-hydroxysuccinimide (NHS) ester-dense rings that are negative for this basal complex marker. (**b**) Parasites prepared by U-ExM where the cytostome marker Kelch13 was conjugated to GFP (K13-GFP) and stained with anti-GFP. Image shows co-localization between K13-GFP and the putative cytostome. NHS ester-dense

*Figure 7 continued on next page*

*Figure 7 continued*

ring. (**c**) Comparison between paraformaldehyde (PFA)-only and PFA-glutaraldehyde fixed U-ExM parasites, showing lysed (PFA only, orange) and intact (PFA-glutaraldehyde, magenta) red blood cell (RBC) membranes. (**d**) In PFA-only fixed parasites, only the cytostomal collar is preserved, while both the collar and bulb are preserved upon PFA-glutaraldehyde fixation. (**e**) K13-GFP parasites were either fixed in PFA only or PFA-glutaraldehyde, prepared by U-ExM, stained with NHS ester (grayscale), SYTOX (cyan), and anti-GFP (cytostome) (magenta) antibodies and imaged using Airyscan microscopy across the asexual blood stage. Zoomed regions show cytostomes. Images are maximum-intensity projections, number on image = Z-axis thickness of projection in μm. White scale bars = 2 μm, yellow scale bars = 500 nm.

The online version of this article includes the following figure supplement(s) for figure 7:

**Figure supplement 1.** Observed cytostome morphologies.

---

(*Birnbaum et al., 2017*). Investigation using this parasite line revealed that the NHS ester-dense ring also stained with K13, suggesting that this structure is a cytostome (*Figure 7b*).

## PFA-glutaraldehyde fixation allows visualization of cytostome bulb

The cytostome can be divided into two main components: the collar, a protein-dense ring at the PPM where K13 is located, and the bulb, a membrane invagination containing RBC cytoplasm (*Milani et al., 2015*; *Xie et al., 2020*). While we could identify the cytostomal collar by K13 staining, these cytostomal collars were not attached to a membranous invagination. Fixation using 4% v/v PFA is known to result in the permeabilization of the RBC membrane and loss of its cytoplasmic contents (*Tonkin et al., 2004*). Topologically, the cytostome is contiguous with the RBC cytoplasm, and so we hypothesized that PFA fixation was resulting in the loss of cytostomal contents and obscuring of the bulb. PFA-glutaraldehyde fixation has been shown to better preserve the RBC cytoplasm (*Tonkin et al., 2004*). Comparing PFA only with PFA-glutaraldehyde fixed parasites, we could clearly observe that the addition of glutaraldehyde preserves both the RBC membrane and RBC cytoplasmic contents (*Figure 7c*). Further, while only cytostomal collars could be observed with PFA-only fixation, large membrane invaginations (cytostomal bulbs) were observed with PFA-glutaraldehyde fixation (*Figure 7d*). Cytostomal bulbs were often much longer and more elaborate spreading through much of the parasite (*Video 1*), but these images are visually complex and difficult to project, so images displayed in *Figure 7* show relatively smaller cytostomal bulbs. Collectively, this data supports the hypothesis that these NHS ester-dense rings are indeed cytostomes and that endocytosis can be studied using U-ExM, but PFA-glutaraldehyde fixation is required to maintain cytostome bulb integrity.

We subsequently harvested K13-GFP parasites across the parasite lifecycle and imaged them following either PFA only or PFA-glutaraldehyde fixation. K13-stained cytostomes were detected at all stages of the parasite lifecycle (*Figure 7e*). Ring-stage parasites typically contained one or two cytostomes, which increased in number during the trophozoite stage and schizogony (*Figure 7e*).

Single cytostomes appear in the area containing the IMC near the apical organelles at the same time as the basal complex forms a complete ring. Cytostomes remain within the IMC area but change positions within the nascent merozoite as segmentation progresses (*Figure 4b*, white asterisks). The majority of merozoites in C1-arrested schizonts contained a single cytostome. This suggests that cytostomes are incorporated into the IMC of merozoites and inherited early

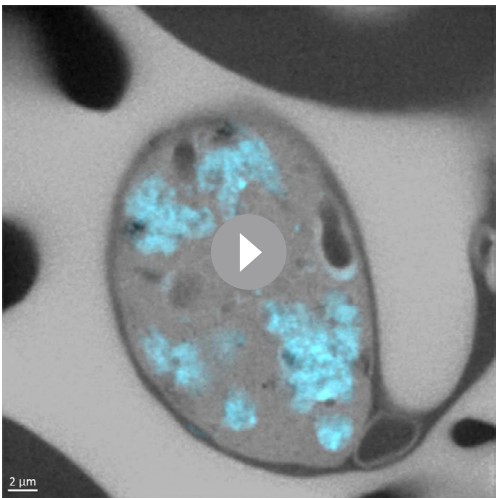

**Video 1.** Slice-by-slice view of paraformaldehyde/ glutaraldehyde fixed parasite. Schizont stage parasite fixed with paraformaldehyde and glutaraldehyde to preserve erythrocyte membrane, prepared using ultrastructural expansion microscopy (U-ExM) and stained with N-hydroxysuccinimide (NHS) ester (protein density, grayscale) and SYTOX (DNA, cyan). This slice-by slice video through the parasite shows the elongated, protein-dense, cytostome invaginations in the parasite.
https://elifesciences.org/articles/88088/figures#video1

in segmentation. Clusters of cytostomes that had not been incorporated into merozoites during segmentation were observed either adjacent to nascent merozoites or as part of the residual body (*Figure 7—figure supplement 1* and *Figure 4b*, yellow asterisk). It is currently unclear whether there are any functional differences between the cytostomes that are incorporated into merozoites and those that are left behind.

## Non-canonical cytostome collar morphologies

We noticed a number of different cytostome morphologies and organizational patterns (*Figure 7—figure supplement 1*). Cytostomes frequently clustered together and did not appear randomly distributed across the PPM (*Figure 7e*). Some cytostomes would form what appeared to be higher order structures where two or three distinct cytostomal collars appeared to be stacked end-on-end (*Figure 7—figure supplement 1*). Cytostomes have a relatively well-defined and consistent size (*Aikawa, 1971*; *Yang et al., 2019*), but occasionally we observed very large cytostomal collars that were approximately twice the diameter of other cytostome collars in the same cell (*Figure 7—figure supplement 1*). It is unclear what the function of these higher order structures or large cytostomes is, if they represent biogenesis transition states, or indeed if they are performing some specialized endocytosis.

## The rhoptries

To invade host RBCs, merozoites secrete proteins from specialized secretory organelles known as the rhoptries and micronemes. While both the rhoptries and micronemes are well studied in the context of *Plasmodium* biology, neither have been investigated in detail using expansion microscopy in *Plasmodium*. We previously showed that fully formed rhoptries can be observed by NHS ester staining alone (*Liffner and Absalon, 2021*), but did not investigate their biogenesis.

## Rhoptries can be observed from early in their biogenesis using NHS ester staining

Rhoptries consist of a neck and bulb region, with the tip of the neck being loaded into the APRs of merozoites. Despite both being formed from Golgi-derived cargo, the neck and bulb regions have distinct proteomes (*Counihan et al., 2013*). We first tracked rhoptry bulb biogenesis across schizogony using antibodies directed against the rhoptry bulb marker rhoptry-associated protein 1 (RAP1).

Nascent rhoptries were detected early in schizogony, with RAP1 foci appearing adjacent to all branches of the outer CP from parasites with 6–10 nuclei in a one-to-one ratio, as described above (*Figure 8a*, *Figure 8—figure supplement 1b*). These foci co-localized with NHS ester densities of the same size and round shape, no elongated neck-like structures were visible by NHS ester (neck biogenesis described in more detail below). This matches reports that rhoptry bulb biogenesis occurs first (*Bannister et al., 2000*; *Counihan et al., 2013*), with neck biogenesis not occurring until segmentation. As early as the last mitotic event during early segmentation, rhoptry bulbs were observed as pairs, with 88 of 93 (95%) CPs observed forming a mitotic spindle being associated with two RAP1-positive NHS ester densities per outer CP branch. Finally, in newly invaded ring-stage parasites, strong RAP1 staining was observed at the PPM/PVM (*Figure 8—figure supplement 1a*), supporting previously reported observations that secreted RAP1 coats the merozoite during invasion (*Riglar et al., 2011*).

Our data not only suggests that rhoptry biogenesis occurs well before segmentation, when nuclei still have several rounds of mitosis to complete, but also that rhoptries remain CP-associated during these mitotic events. Instances of this association with the CP have been observed before (*Bannister et al., 2000*; *Rudlaff et al., 2020*) but its mechanism remains unknown. To our knowledge, this is the first in-depth documentation of a rhoptry–CP association throughout schizogony in *Plasmodium*.

## Rhoptry heterogeneity during early schizogony and segmentation

Rhoptries associated with the same CP during early schizogony sometimes differ in size (*Figure 8—figure supplement 1b*). This is not surprising given the speed of mitotic events requires near-constant biogenesis of new rhoptry bulbs. By the time segmentation is underway, instead of inheriting one sister rhoptry in the final mitotic event of schizogony, each CP will inherit a pair of rhoptries each. At this point, the speed of these mitotic events slows and parasites reach a point of semi-synchrony.

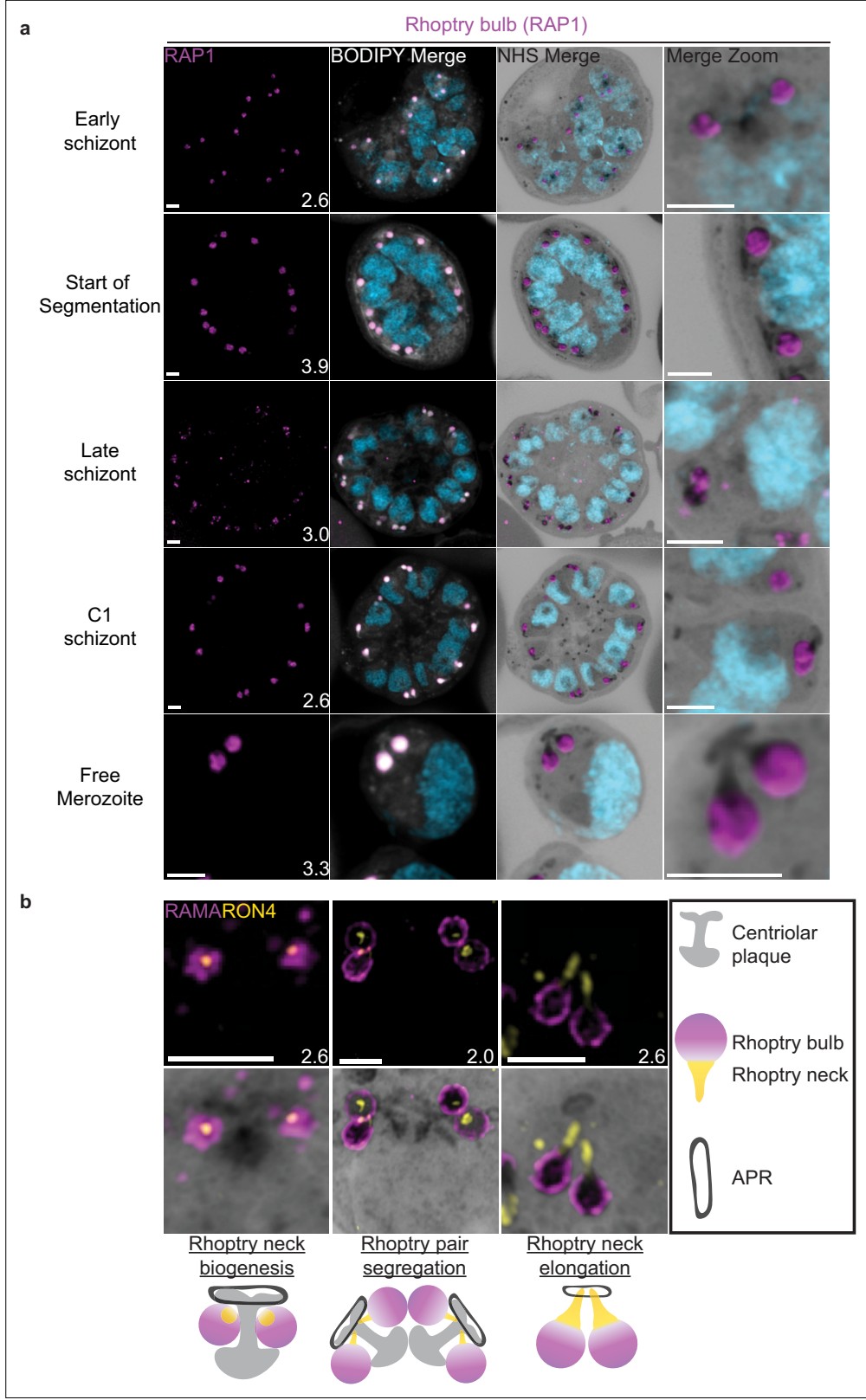

**Figure 8.** Rhoptries undergo biogenesis near the centriolar plaque and are segregated during nuclear division. 3D7 parasites were prepared by ultrastructural expansion microscopy (U-ExM), stained with N-hydroxysuccinimide (NHS) ester (grayscale), BODIPY TRc (white), SYTOX (cyan), and an anti-rhoptry antibodies and imaged using Airyscan microscopy. (**a**) Images of whole parasites throughout schizogony stained using an anti-RAP1 (rhoptry

*Figure 8 continued on next page*

*Figure 8 continued*

bulb; magenta) antibody. (**b**) Zoom into rhoptry pairs of 3D7 parasites that were prepared for U-ExM and stained with NHS ester (grayscale) along with antibodies against RAMA (rhoptry bulb; magenta) and RON4 (rhoptry neck; yellow) to assess rhoptry neck biogenesis. We observed that the rhoptry neck begins as a single focus inside each rhoptry. Rhoptries then get duplicated and segregated alongside the centriolar plaque. During the final mitosis, the rhoptry neck begins to elongate and the rhoptries separate from centriolar plaque. Images are maximum-intensity projections, number on image = Z-axis thickness of projection in μm. Scale bars = 2 μm.

The online version of this article includes the following figure supplement(s) for figure 8:

**Figure supplement 1.** Rhoptry biogenesis during schizogony.

To our surprise, this synchrony does not extend to rhoptry pairs; the two rhoptries inherited by segmenting daughter cells remain different from each other. This heterogeneity in rhoptry pairs during early segmentation has been documented before by electron microscopy (*Bannister et al., 2000*; *Rudlaff et al., 2020*). Of 109 rhoptry pairs imaged in early segmentation schizonts undergoing their last miotic event (where CPs were observed forming mitotic spindles), only 4% had two rhoptries of similar size and density (*Figure 8—figure supplement 1d*). We observed that 40% of these 109 rhoptry pairs had different size but equal NHS ester density, 21% had the same size but different NHS ester density, and 35% differed in both size and NHS ester density (*Figure 8—figure supplement 1e*). As expected from previous reports, this heterogeneity was lost after the completion of this last miotic event and rhoptry neck elongation. Of 98 rhoptry pairs imaged in non-mitotic segmenting parasites, 76% had two rhoptries of similar size and density (*Figure 8—figure supplement 1d*). It is still unclear how the one-to-two rhoptry transition occurs and whether rhoptry heterogeneity has a biological role in biogenesis and maturation of the organelles.

Overall, we present three main observations suggesting that rhoptry pairs undergo sequential de novo biogenesis rather than dividing from a single precursor rhoptry. First, the tight correlation between rhoptry and outer CP branch number suggests that either rhoptry division happens so fast that transition states are not observable with these methods or that each rhoptry forms de novo and such transition states do not exist. Second, the heterogeneity in rhoptry size throughout schizogony favors a model of de novo biogenesis given that it would be unusual for a single rhoptry to divide into two rhoptries of different sizes. Lastly, well-documented heterogeneity in rhoptry density suggests that, at least during early segmentation, rhoptries have different compositions. Heterogeneity in rhoptry contents would be difficult to achieve so quickly after biogenesis if they formed through fission of a precursor rhoptry. While constant de novo biogenesis could explain why one rhoptry can appear smaller or less mature than the other (*Bannister et al., 2000*; *Rudlaff et al., 2020*), it is currently unclear why heterogeneity in rhoptry density only appears during early segmentation and not earlier. Thus, this model is not enough to explain all the variation in rhoptry size and density observed throughout schizogony. Furthermore, a lot of unknowns remain about what exactly governs rhoptry number during the rapid rounds of asynchronous nuclear division (*Klaus et al., 2022*), how the transition to a rhoptry pair is signaled, and how many rounds of de novo rhoptry formation parasites undergo.

## Rhoptry neck biogenesis and elongation

In order to observe rhoptry neck biogenesis in more detail, we stained parasites against the rhoptry apical membrane antigen (RAMA, a rhoptry bulb marker) and rhoptry neck protein 4 (RON4, a rhoptry neck marker) (*Richard et al., 2010*; *Topolska et al., 2004*). RAMA is anchored to the rhoptry bulb membrane and only stains the periphery of the rhoptry bulb as marked by NHS ester (*Figure 8b*, *Figure 8—figure supplement 1c*). RON4 is absent from the earliest rhoptry bulbs, appearing as a focus within the rhoptry bulb shortly before early segmentation and before the rhoptry neck could be distinguished from the bulb by NHS ester staining alone (*Figure 8b*). During early segmentation, when rhoptry pairs first become visible, we observe an uneven distribution of RON4 within each pair. RON4 preferentially associates with one of the rhoptries, with the staining on the second rhoptry being fainter, more diffuse, or even absent in some cases. Of 84 rhoptry pairs observed at this stage, 72 (86%) showed an uneven distribution of RON4. To our surprise, when these rhoptry pairs were of different NHS ester densities, the larger share of RON4 associated with the less dense rhoptry (*Figure 8b*). Previous observations of rhoptry density differences by electron microscopy have been

ascribed to differences in rhoptry age or maturity, with the denser rhoptry being more mature. So, finding RON4 to be more abundant in the less dense rhoptry suggests that either heterogeneous RON4 accumulation cannot be explained by rhoptry age or that the less dense rhoptry is instead the older rhoptry. The RON4-positive rhoptry neck elongates during segmentation, attaining its characteristic shape by mid to late segmentation and becoming observable by both RON4 staining and NHS ester (*Figure 8*). At this point, nearly all rhoptry necks had an equal distribution of RON4 (of 76 rhoptry pairs observed at these stages, 72 [95%] had an equal distribution of RON4).

### The micronemes

Previous studies have suggested that micronemes may be heterogeneous and that apical membrane antigen 1 (AMA1) and other micronemal markers such as erythrocyte binding antigen-175 (EBA175) reside in different subsets of micronemes (*Absalon et al., 2018*; *Ebrahimzadeh et al., 2019*; *Healer et al., 2002*). We reasoned that individual micronemes may be visible using U-ExM and imaged parasites stained with AMA1 and EBA175 to observe their biogenesis and relative distribution. The first microneme marker to appear during schizogony was AMA1. Large puncta of AMA1 appear near the rhoptries when the basal complex is at its maximum diameter (*Figure 9a*). At this point, EBA175 is not yet detectable above background fluorescence. At the end of segmentation, we observe AMA1 has arranged itself into small, densely arranged puncta below the APR and around the rhoptry neck. We also observe EBA175 staining in puncta that are less densely arranged and have little co-localization with AMA1. EBA175 puncta are basal to the AMA1 puncta, being closer to the rhoptry bulb. They also form a cloud of larger diameter than the one formed by AMA1 such that, when viewed from above the APR, two concentric clouds are observed with the core being AMA1 positive and the periphery being EBA175 positive (*Figure 9b and c*). We could observe a punctate NHS ester staining pattern at the apical end of merozoites (*Figure 9b*), which we reasoned could be micronemes. The AMA1 and EBA175 staining we observed in late-stage schizonts partially overlaps with this punctate NHS ester pattern, suggesting that NHS ester punctae are micronemes (*Figure 9c*). However, many NHS ester-positive foci did not stain with either AMA1 or EBA175 despite being morphologically indistinguishable from those which did. This suggests that NHS ester stains more than just the micronemes and that some of these foci may be exonemes, dense granules, or other apical vesicles. Alternatively, it is also possible that these additional NHS ester-positive foci represent micronemes that lack both AMA1 and EBA175.

Using the protease inhibitor E64, we arrested parasites 'post-egress' such that AMA1 was translocated. E64 allows for normal daughter cell maturation but prevents RBC plasma membrane rupture once segmentation is complete (*Hale et al., 2017*). We observe that once AMA1 is translocated, the density of apical AMA1 decreases. EBA175, which does not translocate, increases in density, and moves apically, taking the space AMA1 occupied prior to translocation (*Figure 9c and d*). This is consistent with existing models of sequential microneme translocation and microneme fusion near the APR (*Dubois and Soldati-Favre, 2019*).

### Discussion

In this work, we apply U-ExM to the asexual blood stage of *P. falciparum* to provide new insights into the role of the CP in establishing apical–basal polarity and in coordinating organelle segregation during schizogony (*Figure 10*). In the process, we demonstrate U-ExM can be used to study the biogenesis and protein distribution of a variety of organelles and structures within the parasite. Globally, our observations suggest that the CP is involved in establishing apical–basal polarity within the parasite and that this polarity is established early in schizogony.

We show that the branches of the outer CP appear to act as physical 'tethers,' connecting the nucleus to PPM. This creates a space between the PPM and nuclear envelope where we observe multiple parasite structures and organelles including the Golgi, basal complex, rhoptries, and APRs. These structures and organelles remain CP-associated despite the constant movement and segregation of CPs during mitotic events. In other systems, MTOC or centrosome positioning has been described as a source of cellular polarity, polarizing a cell by directing cargo of secretory organelles to a defined area (*Cowan and Hyman, 2004*; *de Anda et al., 2005*; *Huse, 2012*). While this has not been previously described in *Plasmodium*, this model fits well with our observation of the Golgi being

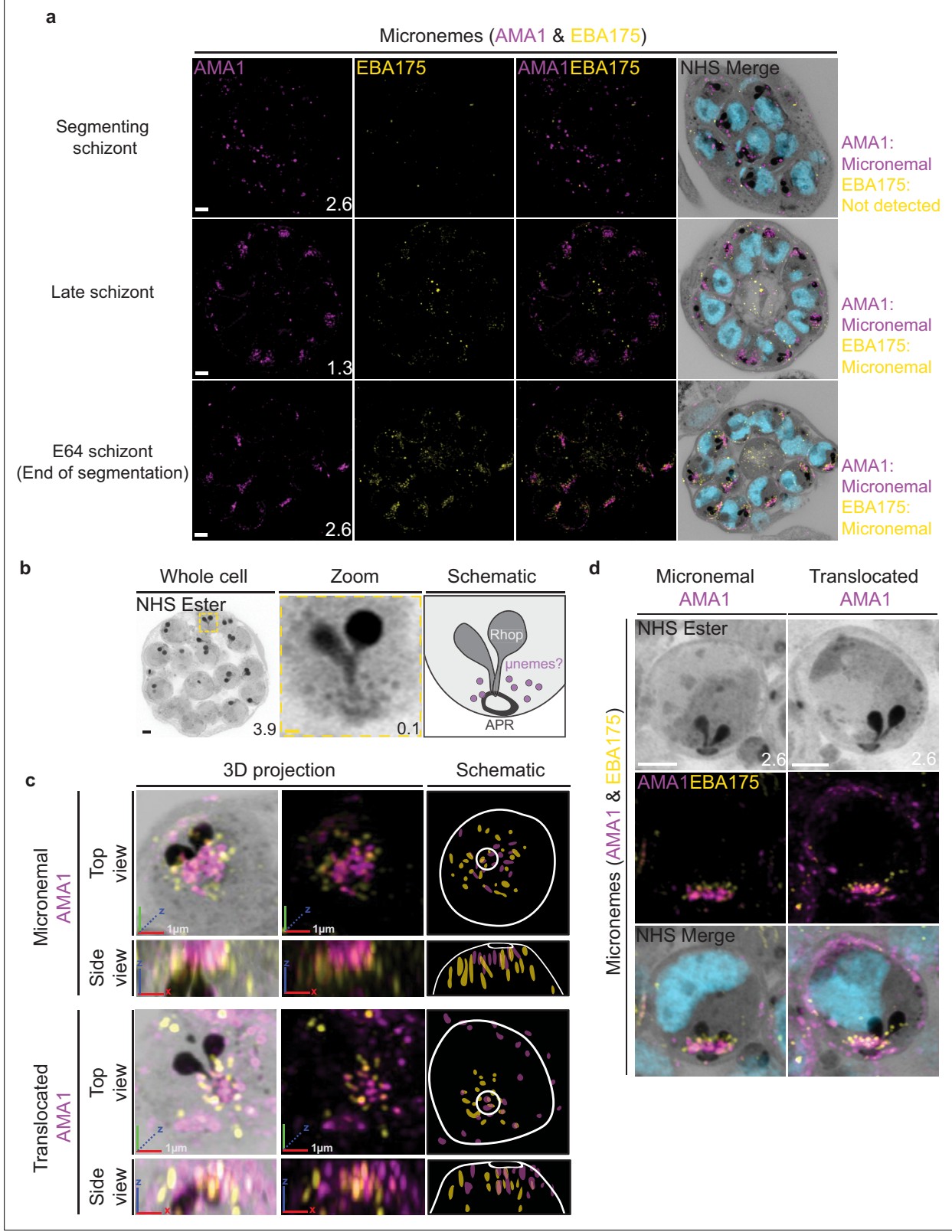

**Figure 9.** Micronemal proteins AMA1 and EBA175 reside in separate micronemes. 3D7 parasites were prepared by ultrastructural expansion microscopy (U-ExM), stained with N-hydroxysuccinimide (NHS) ester (grayscale), SYTOX (cyan), and antibodies against the micronemal markers AMA1 (magenta) and EBA-175 (yellow), and imaged using Airyscan microscopy in segmenting schizonts. (**a**) Images of whole parasites throughout schizogony. In schizonts still undergoing segmentation, AMA1 localized to the apical end while EBA175 was not detected. In late schizonts, both EBA175 and AMA1

*Figure 9 continued on next page*

*Figure 9 continued*

were present in the micronemes. In E64-arrested schizonts, AMA1 was translocated to the merozoite surface while EBA175 remained micronemal. 3D rendering (**b**) and zooms (**c**) of merozoites with either micronemal or translocated AMA1. Foci of AMA1 and EBA175 to not routinely co-localize with each other. Images are maximum-intensity projections, number on image = Z-axis thickness of projection in µm. White scale bars = 2 µm, RGB scale bars for 3D rendering = 1 µm.

adjacent to the CP throughout schizogony. This Golgi positioning could grant local control over the biogenesis of rhoptries, micronemes, dense granules, and IMC, which are all at least partially formed by Golgi-derived cargo (*Counihan et al., 2013*; *Dubois and Soldati-Favre, 2019*; *Griffith et al., 2022*; *Sloves et al., 2012*). While the biogenesis of the APRs seems to also be coupled to the CP, we do not yet know how they are nucleated or whether they are also dependent on Golgi-derived cargo.

This study also shows CP-coupled organelle segregation and biogenesis to an extent never observed before. Specifically, we observe that the Golgi, rhoptries, and basal complex match the branches of the outer CP in number throughout early schizogony and segregate with CPs during the mitotic events of schizogony. As this study used fixed cells, we lack the temporal power to precisely define the order of these events. Using known markers of parasite age, however, we describe putative transition states that nascent rhoptries and basal complexes adopt when being segregated with CPs. We put forward a model where outer CP-associated events occur before inner CP-associated events. That is, the duplication of the outer CP and organelles precedes inner CP duplication and the formation of a mitotic spindle such that this cytoplasmic duplication represents the first identifiable step in the commitment of a nucleus to the next round of mitosis. Interestingly, nuclei can sometimes have CPs forming a mitotic spindle that have already committed to the next round of mitosis by duplicating their cytoplasmic cargo. That is, a mitotic nucleus can show four outer CP branches and sets of apical organelles at the same time. While this supports our hypothesis that duplication of apical organelles and the outer CP components happens upstream from intranuclear mitotic events, it does not tell us how this relates to the genome copy number inside the nucleus. So, we cannot say whether DNA replication and associated checkpoints occur upstream or downstream from the cytoplasmic events that seem to commit a CP to mitosis.

Lastly, we contribute important evidence toward hypotheses across several open questions regarding organelle biogenesis and segregation in *Plasmodium*. We observe contacts between the outer CP and both the mitochondria and apicoplast during fission that suggest a role for this structure in monitoring copy number of these organelles. We also document sequential fission events in both the apicoplast and mitochondria that are suggestive of branching point fission. Lastly, we see temporal and spatial localization patterns of AMA1, EBA175, and RON4 that support theories of heterogeneity within the apical organelles.

U-ExM represents an affordable and adaptable sample preparation method that can be applied to any microscope to produce images with far greater visible detail than conventional light microscopy platforms. Applying this technique to the visualization of organelle biogenesis and segregation throughout schizogony allowed us to observe these processes in the context of structures that could previously only be investigated in detail using electron microscopy. The flexibility and scalability of this technique allowed us to image more than 600 individual parasites at a variety of developmental stages, increasing the confidence of our observations and giving them some temporal resolution. To our knowledge, this article represents the most comprehensive study of a single organism using U-ExM, with a total of 13 different subcellular structures investigated.

Our inability to pinpoint the nucleation site of the SPMTs or resolve the plasma membrane from the IMC highlights some of the limitations of U-ExM as applied in this work. The subpellicular network that holds IMC1g and lines the cytoplasmic face of the IMC sits approximately 20 nm below the parasite surface, where we would find MSP1 (*Kudryashev et al., 2012*). Thus, the distance between IMC1g and MSP1 post-expansion is around 90 nm. This is below our imaging resolution with Airyscan 2 and close to the maximum resolution we could achieve through other super-resolution methods compatible with our current setup when antibody effects are considered. Thus, some parasite structures remain beyond the resolution achieved in this study. In order to resolve the IMC from the plasma membrane or resolve multiple APRs using light microscopy, we would need to employ single-molecule localization microscopy or iterative expansion microscopy (*Louvel et al., 2022*), which increases expansion factor to ~20×.

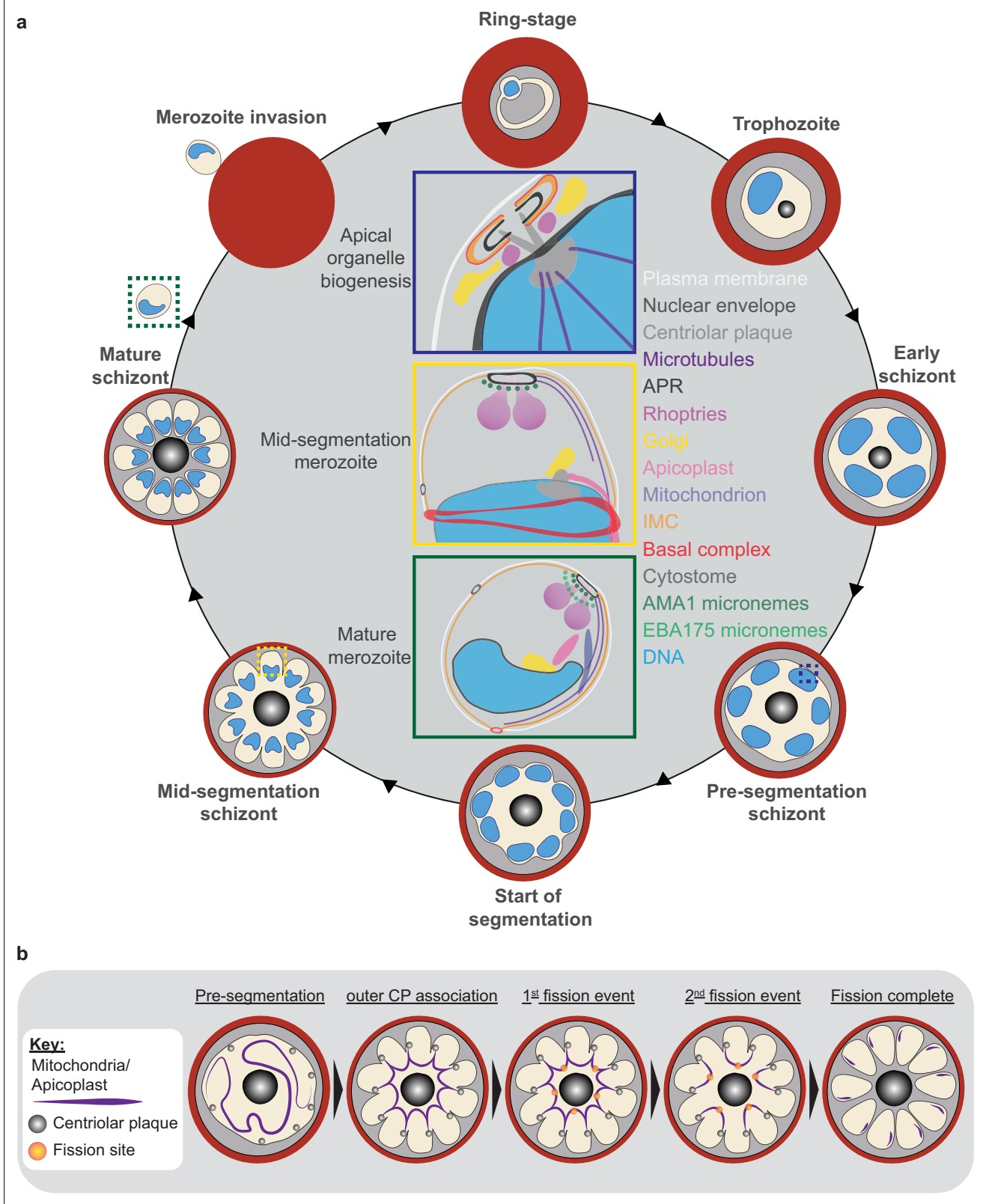

**Figure 10.** Summary of organelle organization and fission during schizogony. (**a**) Apical organelle biogenesis: Biogenesis of the rhoptries, Golgi, basal complex, and apical polar rings occur at outer centriolar plaque (CP), between the nuclear envelope and parasite plasma membrane. Duplication and segregation of these organelles appears to be tied to CP duplication and segregation following nuclear division. Mid-segmentation merozoite: the rhoptry neck is distinguishable from the bulb, and AMA1-positive micronemes are present at the apical end of the forming merozoite. Each merozoite

*Figure 10 continued on next page*

*Figure 10 continued*

has inherited a cytostome. Subpellicular microtubules stretch the entire distance from the apical polar rings and the basal complex. The apicoplast has attached to the outer CP and begun fission. Mature merozoite: the parasite has completed segmentation, and each merozoite contains a full suite of organelles. The CP is no longer visible, and EBA175-positive micronemes are both visible and separate from AMA1-positive micronemes. (**b**) Model for fission of the mitochondrion and apicoplast. Prior to fission, both the apicoplast and mitochondrion branch throughout the parasite cytoplasm, before associating with the outer CP of each forming merozoite. For the apicoplast, this occurs during the final mitosis, but not until late in segmentation for the mitochondrion. Following outer CP association, the apicoplast and mitochondrion undergo a first fission event, which leaves an apicoplast and mitochondrion shared between forming merozoite pairs. Subsequently, both organelles undergo a second fission event, leaving each forming merozoite with a single apicoplast and mitochondrion.

We also noticed some drawbacks and artifacts introduced by U-ExM. Most visually striking was that the hemozoin crystal of the food vacuole does not expand (*Coronado et al., 2014*), which leaves a large space that lacks NHS ester staining. For nearly all antibodies used in this study, significant off-target fluorescence was observed inside the food vacuole. Thus, U-ExM may not be as useful for studying food vacuole biology. Occasionally, significant SYTOX (DNA stain) fluorescence was observed at either the nuclear envelope or PPM. It is unclear if this represents an expansion-induced artifact or a PFA-fixation artifact that is only now observable. Lastly, for cells stained with MitoTracker, some non-specific background was observed that seemed to correlate with protein density as observed by NHS ester.

Malaria parasites have been extensively studied using electron microscopy to determine their ultra-structure and live-cell microscopy to observe their most dynamic processes in real time. Much of what we uncovered in this study involved dynamic processes that are too small to be resolved using conventional live-cell microscopy. Specifically, we made important observations about *P. falciparum* organelle biogenesis and the organization of *Plasmodium* cell division around the CP. Rather than a replacement for any existing microscopy techniques, we see U-ExM as a complement to the suite of techniques available to study the cell biology of malaria parasites, which bridges some of the limitations of electron microscopy and live-cell microscopy. As such, there are many parasite processes that are logical candidates for investigation by U-ExM. Some of these have been highlighted in this article, but others remain completely unexplored. Merozoite invasion, for example, has been well-studied using a variety of microscopy techniques (*Geoghegan et al., 2021*; *Hanssen et al., 2013*; *Liffner et al., 2022*; *Riglar et al., 2011*; *Weiss et al., 2015*), but U-ExM would allow us to visualize how all the apical organelles associate with each other and rearrange in three dimensions across the established time course (*Weiss et al., 2015*) of this process. Another logical candidate for investigation by U-ExM is the rapid disassembly of the IMC and other merozoite organelles immediately following invasion (*Ferreira et al., 2020*), a process where the parasite undergoes rapid and dramatic morphological rearrangements that lie beyond the resolution of live-cell microscopy.

## Materials and methods

### *Plasmodium falciparum* culture

Unless otherwise stated, all parasites in this study were 3D7-Cas9 (*Rudlaff et al., 2019*). For imaging of the apicoplast, the previously generated ACP-transit-peptide-GFP cell line was used (*Florentin et al., 2020*). For imaging of Kelch13, the previously generated 2xFKBP-GFP-K13 parasites were used (*Birnbaum et al., 2017*). For imaging of the basal complex, the previously generated CINCH-smV5 cell line was used (*Rudlaff et al., 2019*).

Identity of 3D7-Cas9 parasites was confirmed by whole-genome sequencing. Identity of transgenic parasites was confirmed by resistance to drug selectable markers, expression of fluorescently tagged proteins, and PCR. All parasite lines were tested periodically for *Mycoplasma* contamination.

All parasites were cultured in O$^+$ human RBCs at 4% hematocrit in RPMI-1640 containing 25 mM HEPES, 50 mg/L hypoxanthine, 0.21% sodium bicarbonate, and 0.5% w/v Albumax II (*Trager and Jensen, 1976*). All parasite cultures were incubated on a shaker at 37°C in a gas mixture of 1% O$_2$, 5% CO$_2$, and 94% N$_2$ as previously described. The smHA-tagged Pf3D7_0311800 (ATP Synthase F0 Subunit D) cell line was maintained under selection of 5 nm WR99210. Apicoplast targeting signal-GFP line was maintained under selection of 2.5 ug/ml of Blasticidin-hcl. 2xFKBP-GFP-K13 parasites were maintained under selection of 0.9 μm DSM1.

**Table 1.** Oligonucleotides for cloning and integration PCR.

| Oligo/gBlock name | Sequence (5'→3') |
| --- | --- |
| JDD44 | TGGGGTGATGATAAAATGAAAG |
| JDD56 | ACACTTTATGCTTCCGGCTCGTATGTTGTG |
| JDD4889 | TATTGTCAAATCGTTACCTCTATG |
| JDD4890 | AAACCATAGAGGTAACGATTTGAC |
| JDD4891 | TAGgcggccgcGGTCCTACACCAATAAATATCA |
| JDD4892 | GTCTGATTCTTCCCATCaggccttccggaccgcggGGTCCCTTCATTGTAGACTTTTTATTATTGAAC |
| JDD4893 | GTCTACAATGAAGGGACCCccgcggtccggaaggcctGATGGGAAGAATCAGACAAATGGT |
| JDD4894 | GATctcgagcAGcGGcAAtGAcTTcACgAATTCTCTTATTTCTTGTTTTTGCATTTCCT |
| SAB257 | CGGACCGAGAATTTATGTCCATTAACGTC |
| SAB471 | TGGTATTAATGGATGAAGACACACA |
| SAB472 | GTAATGGAATAGCTTTATATATGTACCTTCAT |
| SAB473 | TATGTGATCCATACATACCTGTTCAGAC |

Parasites were routinely synchronized using sorbitol lysis. Briefly, parasite cultures were resuspended in 5% w/v D-sorbitol, resulting in the selective lysis of schizont-stage parasites (*Lambros and Vanderberg, 1979*).

For samples where parasites were arrested as schizonts using either trans-epoxysuccinyl-L-leucylamido(4guanidino)butane (E64) (*Salmon et al., 2001*) or compound 1 (C1) (*Taylor et al., 2010*), late schizont-stage cultures were treated with either 10 µm E64 for ~3 hr or 5 µm C1 for ~5 hr.

## Plasmid generation and transfection

For imaging of the mitochondria, a cell line where ATP-Synthase F0 Subunit D (Pf3D7_0311800) had a C-terminal spaghetti-monster HA tag was generated (*Figure 5—figure supplement 2*). To create the Pf3D7_0311800 smHA HDR plasmid, the 3D7_0311800 5' and 3' homology regions were PCR amplified from 3D7 genomic DNA with oligonucleotides oJDD4893/oJDD4894 and oJDD4891/oJDD4892, respectively. The two pieces were fused together using Sequence Overlap Extension PCR (SOE PCR) using oJDD4891/oJDD4894 and the piece was digested with NotI/XhoI and ligated with T4 ligase to generate pSAB55. To create the PF3D7_0311800 targeting guide RNA plasmid, oJDD4889/oJDD4890 were annealed, phosphorylated, and ligated into BpiI-digested pRR216 to generate pSAB81. All oligonucleotide sequences are shown in *Table 1*.

For transfection, 100 µg of pSAB55 plasmid was linearized with StuI and transfected into 3D7-Cas9, along with 100 µg of pSAB81. A day following transfection, parasites were treated with 5 nm WR99210 until 13 days, when resistant parasites were detected.

## Ultrastructure expansion microscopy

U-ExM was performed as previously described with minor modification (*Bertiaux et al., 2021*; *Gambarotto et al., 2019*; *Liffner and Absalon, 2021*). Then, 12 mm round Coverslips (Fisher, Cat# NC1129240) were treated with poly-D-lysine for 1 hr at 37°C, washed twice with MilliQ water, and placed in the wells of a 12-well plate. Parasite cultures were set to 0.5% hematocrit, and 1 mL of parasite culture was added to the well containing the coverslip for 15 min at 37°C. Culture supernatants were removed, and cultures were fixed with 1 mL of 4% v/v PFA in 1× PBS for 15 min at 37°C. For some experiments visualizing cytostomes, cultures were instead fixed in 4% v/v PFA + 0.01% v/v glutaraldehyde in 1× PBS. Following fixation, coverslips were washed three times with 37°C PBS before being treated with 1 mL of 1.4 % v/v formaldehyde/2% v/v acrylamide (FA/AA) in PBS. Samples were then incubated at 37°C overnight.

Monomer solution (19% w/w sodium acrylate [Sigma, Cat# 408220], 10% v/v acrylamide [Sigma, Cat# A4058, St. Louis, MO], 0.1% v/v N,N'-methylenebisacrylamide [Sigma, Cat# M1533] in PBS) was typically made the night before gelation and stored at −20°C overnight. Prior to gelation, FA/

AA solution was removed from coverslips, and they were washed once in PBS. For gelation, 5 µL of 10% v/v tetraethylenediamine (TEMED; Thermo Fisher, Cat# 17919) and 5 µL of 10% w/v ammonium persulfate (APS; Thermo Fisher, Cat# 17874) were added to 90 µL of monomer solution and briefly vortexed. Subsequently, 35 µL was pipetted onto parafilm and coverslips were placed (cell side down) on top. Gels were incubated at 37°C for 30 min before being transferred to wells of a 6-well plate containing denaturation buffer (200 mM sodium dodecyl sulfate [SDS], 200 mM NaCl, 50 mM Tris, pH 9). Gels were incubated in denaturation buffer with shaking for 15 min, before separated gels were transferred to 1.5 mL tubes containing denaturation buffer. 1.5 mL tubes were incubated at 95°C for 90 min. Following denaturation, gels were transferred to 10 cm Petri dishes containing 25 mL of MilliQ water for the first round of expansion and placed onto a shaker for 30 min three times, changing water in between. Gels were subsequently shrunk with two 15 min washes in 25 mL of 1× PBS, before being transferred to 6-well plates for 30 min of blocking in 3% BSA-PBS at room temperature. After blocking, gels were incubated with primary antibodies, diluted in 3% BSA-PBS, overnight. After primary antibody incubation, gels were washed three times in 0.5% v/v PBS-Tween 20 for 10 min before incubation with secondary antibodies diluted in 1× PBS for 2.5 hr. Following secondary antibody incubation, gels were again washed three times in PBS-Tween 20, before being transferred back to 10 cm Petri dishes for re-expansion with three 30 min MilliQ water incubations.

Gels were either imaged immediately following re-expansion or stored in 0.2% w/v propyl gallate in MilliQ water until imaging. For gels stained with BODIPY TRc, the fully expanded gel was incubated overnight at room temperature in 0.2% w/v propyl gallate (Acos Organics, Cat# 131581000) solution containing BODIPY TRc (2 µM final concentration).

For parasites stained with MitoTracker Orange CMTMRos (Thermo Fisher, M7510), parasite cultures were resuspended in incomplete media (RPMI-1640 containing 25 mM HEPES, 50 mg/L hypoxanthine, and 0.21% sodium bicarbonate) containing 300 nM MitoTracker Orange CMTMRos. Parasite cultures were then stained with MitoTracker for 35 min while settling on poly-D-lysine-coated coverslips. From this point, the expansion protocol was followed as described above, with the exception that all steps when possible were carried out protecting the sample from light.

## Cryopreservation and thawing of gels

A proportion of gels imaged in this study were cryopreserved and subsequently thawed prior to imaging (*Louvel et al., 2022*). Gels were frozen either unstained, following the first round of expansion, or frozen stained, following the second round of expansion. To freeze, a portion of the expanded gel was placed into a 6-well dish and washed three times with 50% glycerol in MilliQ water for 30 min. Fresh glycerol was then added, and the gels were stored at –20°C for future use. To thaw unstained gels, the glycerol was replaced with MilliQ water and incubated at room temperature for 30 min. Gels were then washed and shrunk with three 20 min washes in 1× PBS at room temperature before proceeding with the antibody staining process normally. Stained gels were thawed with three washes in MilliQ water for 30 min before proceeding with imaging as normal.

## Stains and antibodies

A comprehensive list of all stains and antibodies used in this study, their working concentrations, and source(s) is provided in *Table 2*.

## Image acquisition

Immediately before imaging, a small slice of gel ~10 mm × ~10 mm was cut and mounted on an imaging dish (35 mm Cellvis coverslip bottomed dishes NC0409658, Fisher Scientific) coated with poly-D lysine. The side of the gel containing sample is placed face down on the coverslip and a few drops of ddH$_2$0 are added after mounting to prevent gel shrinkage due to dehydration during imaging. All images presented in this study were taken using either a Zeiss LSM800 AxioObserver with an Airyscan detector, or a Zeiss LSM900 AxioObserver with an Airyscan 2 detector. Imaging on both microscopes was conducted using a ×63 Plan-Apochromat objective lens with a numerical aperture of 1.4. All images were acquired as Z-stacks that had an XY pixel size of 0.035 µm and a Z-slice size of 0.13 µm.

**Table 2.** Summary of all antibodies and stains used in this study.

| Primary antibodies | Antibody species | Antibody source (Cat#) | Ab concentration | Reference |
|---|---|---|---|---|
| Anti-alpha tubulin (Clone B-5-1-2) | Mouse (IgG1) | Thermo Fisher (32-2500) | 1:500 | |
| Anti-centrin (Clone 20H5) | Mouse (IgG2a) | Sigma-Aldrich (04-1624) | 1:200 | |
| Anti-*Hs*centrin1 | Rabbit | Thermo Fisher (PA5-29986) | 1:500 | |
| Anti-polyE (IN105) | Rabbit | Adipogen (AG-25B-0030-C050) | 1:500 | |
| Anti-ERD2 (MRA-1) | Rabbit | BEI Resources MR4 | 1:2000 | *Elmendorf and Haldar, 1993* |
| Anti-HA (3F10) | Rat | Roche (12158167001) | 1:50 | |
| Anti-GFP | Rabbit | OriGene (TP401) | 1:2000 | |
| Anti-RAP1 (2.29) | Mouse | European Malaria Reagent Repository | 1:500 | *Hall et al., 1983* |
| Anti-RON4 | Mouse | Gift from Alan Cowman | 1:100 | |
| Anti-RAMA | Rabbit | Gift from Ross Coppel | 1:200 | *Topolska et al., 2004* |
| Anti-AMA1 | Rabbit | Gift from Carole Long | 1:500 | |
| Anti-EBA175 (3D7) | Mouse | Gift from Alan Cowman | 1:500 | *Sim et al., 2011* |
| Anti-Aldolase | Rabbit | Abcam (ab207494) | 1:2000 | |
| Anti-Histone H3 | Rabbit | Abcam (ab1791) | 1:1000 | |
| Anti-BIP | Rabbit | Generated by Dvorin Lab | 1:2000 | |
| Anti-GAP45 | Rabbit | Gift from Julian Rayner | 1:2000 | *Jones et al., 2009* |
| Anti-IMC1g | Rabbit | Generated by Dvorin Lab | 1:1000 | *Cepeda Diaz et al., 2023* |
| Anti-MSP1 (1E1) | Rabbit | Gift from Anthony Holder | 1:250 | *Blackman et al., 1994* |
| Anti-hemoglobin | Rabbit | Thermo Fisher (PA5-102943) | 1:1000 | |

| Secondary antibodies | Antibody species | Antibody source | Antibody concentration |
|---|---|---|---|
| Anti-mouse IgG Alexa Fluor 488 | Goat | Thermo Fisher (A28175) | 1:500 |
| Anti-mouse IgG Alexa Fluor 555 | Goat | Thermo Fisher (A21428) | 1:500 |
| Anti-rabbit IgG Alexa Fluor 488 | Goat | Thermo Fisher (A11034) | 1:500 |
| Anti-rabbit IgG Alexa Fluor 555 | Goat | Thermo Fisher (A21428) | 1:500 |
| Anti-rat IgG Alexa Fluor 488 | Goat | Thermo Fisher (A11006) | 1:500 |
| Anti-mouse IgG2a Alexa Fluor 488 | Goat | Thermo Fisher (A21131) | 1:500 |
| Anti-mouse IgG1 Alexa Fluor 594 | Goat | Thermo Fisher (A21125) | 1:500 |

| Stains | Stain source (Cat#) | Stain concentration |
|---|---|---|
| NHS ester Alexa Fluor 405 | Thermo Fisher (A30000) | 1:250 (8 µM) in DMSO |
| BODIPY TR ceramide | Thermo Fisher (D7549) | 1:500 (2 µM) |
| SYTOX Deep Red | Thermo Fisher (S11381) | 1:1000 (1 µM) in DMSO |

## Image analysis

### Image processing and presentation

All images were Airyscan processed using 3D processing at moderate filter strength on ZEN Blue Version 3.1 (Zeiss, Oberkochen, Germany).

The majority of images presented in this study are presented as maximum-intensity z-projections, but those that contain BODIPY TRc are presented as average intensity projections for viewing and

interpretation purposes. For images containing NHS ester, the gamma value of this channel was set to 0.45 rather than 1 as this allowed discernment of a greater number of parasite structures. It should be noted that because of this the fluorescence intensities shown in the NHS ester channel are not linear.

### 3D rendering

3D renderings of micronemal proteins AMA1 and EBA175 were produced using the 3D analysis package on ZEN Blue version 3.5.

### Measurement of interpolar spindles and SPMTs

All length measurements reported in this study were obtained using the 'Measure 3D distance' function of ZEN Blue version 3.1. The length of interpolar spindle microtubules and SPMTs was determined as the 3D distance between the start and end points of continuously stained stretches of anti-tubulin staining. Interpolar microtubules were defined as those whose staining appeared to contact both CPs as defined by NHS ester, while non-interpolar microtubules were those that did not meet these criteria. SPMTs were only measured in C1-arrested schizonts that had visibly completed segmentation based on the basal complex as visualized by NHS ester. Any microtubule that did not appear connected to the APR, or extend toward the basal complex was excluded from the analysis. Cell diameter was defined as the greatest XY distance on any z-slice between two points of the parasite as defined by NHS ester staining. Merozoite length was defined as the 3D distance between the center of the APRs and basal complex as defined by NHS ester staining.

### Apicoplast and mitochondria area analysis

All area measurements presented were obtained using the 'Area' function on ZEN Blue version 3.1. Images were presented as a maximum-intensity projection before free hand outlining the apicoplast or the mitochondria in each image. The sum of all fragments was then calculated to find the total area of the organelles per cell.

### Mitochondria residual body analysis

In C1-arrested schizonts, the proportion of total mitochondria staining found in the residual body was calculated as follows. Using ZEN Blue version 3.1, a maximum-intensity projection of the entire cell was generated and based on the NHS staining, the entire parasite was defined as the region of interest. Signal intensity of the channel staining the mitochondria was calculated inside the full cell, which was defined as total mitochondria fluorescence. Subsequently, the residual body was defined the area within the parasite vacuole, but external to all merozoite plasma membranes, as based on BODIPY TRc staining. A second maximum-intensity projection of this subsection of the schizont was made, the residual body was defined as the region of interest, and this signal intensity inside this region of interest was defined as residual body mitochondria fluorescence.

To determine residual body mitochondria fluorescence (RB) as % total mitochondria fluorescence (total), the following equation was used:

$$\left( \frac{RB\ fluorescence}{Total\ fluorescence} \right) \times 100$$

In each of the parasites included in this analysis, the number of merozoites was defined as the number of distinct nuclei as determined by SYTOX staining. To determine residual body mitochondria fluorescence as % of one merozoite, the following equation was used:

$$\left( RB\ fluorescence \ \div \ \left( \frac{Total\ fluorescence - RB\ fluorescence}{Number\ of\ merozoites} \right) \right) \times 100$$

Cells where no mitochondria fluorescence was visible inside the residual body, while visible inside merozoites, were defined as having no residual body mitochondria. An attempt was made to do a similar analysis on apicoplast stained cells, but no visible apicoplast staining was ever observed in the residual body.

## Statistical analysis

### Estimation of actual distance from expanded samples

Expansion factors for 43 gels used in this study were determined as follows. Gels were assumed to have an initial diameter of 12 mm as they are formed on a 12-mm-diameter coverslip. Gels were subsequently measured following expansion to the nearest whole millimeter, and the expansion factor was defined as the expanded gel diameter divided by the initial gel diameter (12 mm). Gels whose edges were damaged or malformed, and therefore their diameters could not be actually measured, were excluded. Gels in this study had a median expanded diameter of 51 mm, which corresponds to a median expansion factor of 4.25 (*Figure 1—figure supplement 1*).

### Generation of graphs and statistical analysis

All graphs presented in this study were generated using GraphPad PRISM 9. All error bars in this study represent standard deviation. Differences between samples analyzed by ANOVA was determined as difference where the p-value was <0.05. For scatterplots, slopes were considered significantly non-zero when the p-value was <0.05.

## Data accessibility

Results in this study are underpinned by 647 3D Airyscan images of U-ExM parasites at multiple lifecycle stages with multiple combinations of stains. All images are publicly available through the following data repository: https://doi.org/10.5061/dryad.934mw6mp4.

## Acknowledgements

We thank David Roos, Akhil Vaidya, and Rodolpho Ornitz Oliveira Souza for insightful discussions, Vincent Louvel for U-ExM expertise and sharing the protocol for cryopreservation and thawing of gels, and Taco Kooij, and Julie Verhoef for their help with MitoTracker experiments and critical reading of the manuscript. We thank Tobias Spielmann and Isabelle Henshall for generously sharing K13-GFP parasites, and insightful discussion. We thank Julian Rayner, Alan Cowman, Carole Long, BEI Resources (NIAID, NIH), and the European Malaria Reagent Repository for provision of antibodies. This work was supported by NIH R01 AI145941 (JDD) and F31 AI172110 (AKCD), and the American Heart Association 23POST1011626 (BL).

## Additional information

### Funding

| Funder | Grant reference number | Author |
|---|---|---|
| American Heart Association | 10.58275/aha. 23post1011626.pc.gr. 161347 | Benjamin Liffner |
| National Institutes of Health | R01 AI145941 | Jeffrey D Dvorin |
| National Institutes of Health | F31 AI172110 | Ana Karla Cepeda Diaz |
| NHMRC | GNT2020822 | Danny W Wilson |
| Hospital Research Foundation | C-MCF-52-2019 | Danny W Wilson |

The funders had no role in study design, data collection and interpretation, or the decision to submit the work for publication.

### Author contributions

Benjamin Liffner, Conceptualization, Data curation, Formal analysis, Investigation, Visualization, Methodology, Writing – original draft, Writing – review and editing; Ana Karla Cepeda Diaz,

Conceptualization, Data curation, Formal analysis, Funding acquisition, Investigation, Visualization, Methodology, Writing – original draft, Writing – review and editing; James Blauwkamp, Data curation, Formal analysis, Investigation, Writing – original draft; David Anaguano, Investigation, Writing – review and editing; Sonja Frolich, Methodology; Vasant Muralidharan, Resources, Writing – review and editing; Danny W Wilson, Writing – review and editing; Jeffrey D Dvorin, Conceptualization, Resources, Supervision, Funding acquisition, Project administration, Writing – review and editing; Sabrina Absalon, Conceptualization, Resources, Supervision, Funding acquisition, Methodology, Project administration, Writing – review and editing

### Author ORCIDs
Benjamin Liffner ⓘ https://orcid.org/0000-0002-1573-6139
Ana Karla Cepeda Diaz ⓘ https://orcid.org/0000-0002-2033-461X
James Blauwkamp ⓘ http://orcid.org/0000-0002-7639-2800
Sonja Frolich ⓘ http://orcid.org/0000-0002-8673-3437
Vasant Muralidharan ⓘ http://orcid.org/0000-0001-6367-6284
Danny W Wilson ⓘ https://orcid.org/0000-0002-5073-1405
Jeffrey D Dvorin ⓘ https://orcid.org/0000-0002-5883-7271
Sabrina Absalon ⓘ http://orcid.org/0000-0003-2468-8156

Reviewer #1 (Public Review): https://doi.org/10.7554/eLife.88088.3.sa1
Reviewer #2 (Public Review): https://doi.org/10.7554/eLife.88088.3.sa2
Reviewer #3 (Public Review): https://doi.org/10.7554/eLife.88088.3.sa3
Author Response https://doi.org/10.7554/eLife.88088.3.sa4

## Additional files

### Supplementary files
• MDAR checklist

### Data availability
All images underlying this publication are publically available through Dryad at the following link https://doi.org/10.5061/dryad.9s4mw6mp4.

The following dataset was generated:

| Author(s) | Year | Dataset title | Dataset URL | Database and Identifier |
|---|---|---|---|---|
| Benjamin L, Diaz AKC, Blauwkamp J, Anaguano D, Frölich S, Muralidharan V, Wilson DW, Dvorin J, Absalon S | 2023 | Data from: Atlas of *Plasmodium falciparum* intraerythrocytic development using expansion microscopy | https://doi.org/10.5061/dryad.9s4mw6mp4 | Dryad Digital Repository, 10.5061/dryad.9s4mw6mp4 |

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
