## [Editor Report · eLife assessment]

This **important** study provides an unprecedented overview of the subcellular organization of proliferative blood-stage malaria parasites using expansion microscopy. The localization of multiple parasite organelles is comprehensively probed using three-dimensional super-resolution microscopy throughout the entire intraerythrocytic development cycle. This work provides a **compelling** framework to investigate in future more deeply the unconventional cell biology of malaria-causing parasites.

---

## [Referee Report · Reviewer #1 (Public Review)]

This paper consists of a comprehensive analysis of the malaria parasite *Plasmodium falciparum* during its development in erythrocytes, using expansion microscopy. The authors used general dyes to stain membranes or proteins and a set of specific markers to label diverse cellular structures of the parasite, with a particular focus on the centriolar plaque.

This is by nature a purely descriptive study, providing remarkable images with great details on subcellular structures such as the centriolar plaque, the basal complex, the cytostome and rhoptries. The work is extremely well performed and the images are beautiful. This study confirms a number of previous observations and illustrates the strength of expansion microscopy, an affordable and adaptable sample preparation method that will undoubtedly become standard in the field.

This study provides a valuable resource that can serve as a reference dataset for the analysis of *P. falciparum* and other apicomplexan parasites.

---

## [Referee Report · Reviewer #2 (Public Review)]

In this work the authors describe the shape and interconnectedness of intracellular structures of malaria blood stage parasites by taking advantage of expansion microscopy. Compared to previous microscopy work with these parasites, the strength of this paper lies in the increased resolution and the fact that the NHE ester highlights protein densities. Together with the BodipyC membrane staining, this results in data that is somewhere in between EM and standard fluorescence microscopy: it has higher resolution than standard fluorescence microscopy and provides some points of reference of different cellular structures due to the NHE ester/BodipyC.

This study makes many interesting and useful observations and although it is somewhat "old school descriptory" in its presentation, researchers working in many different areas will find something of interest here. This ranges from mitosis, to organisation and distribution of major cellular structures, endocytosis and invasion, overall providing a rich and interesting resource. The results section is long but by taking the space to explain everything in detail, it has the advantage that it clearly transpires how things were done and on how many cells a conclusion is based on. Further the authors often also included a brief interpretation of their findings with a very open assessment what it does and what it does not show, highlighting interesting questions left by the data.

Overall this is a very nice and useful paper that will be of interest to many, particularly those working on nuclear division, cytokinesis, endocytosis or invasion in malaria parasites. The spatiotemporal arrangement and interconnection of subcellular structures will also give a framework for specific functional studies.

---

## [Referee Report · Reviewer #3 (Public Review)]

In their study the authors analyze the localization of multiple organelles and subcellular structure of blood stage malaria parasites with unprecedented detail. They use a 3D super-resolution imaging technique that has gained popularity in the protozoan field, ultrastructure expansion microscopy. Building on markers and labels established in the field they generate an appealing collection of images for all stages of the intraerythrocytic developmental stages of asexual blood stage parasites with some focus on nuclear division and cell segmentation stages.

The authors have made a very good effort to address all the comments raised by the reviewers providing more clarity to the manuscript and appropriate interpretations of their results. Particularly the sharing of their image data in the Dryad repository adds significant value to their work.

---

## [Author Response]

The following is the authors’ response to the original reviews.

We greatly appreciate the overwhelmingly positive summaries from all three reviewers and the eLife editorial team. All reviewers provided extremely detailed feedback regarding the initially submitted manuscript, we appreciate their efforts in helping us improve this manuscript. Below, are listed each of the specific comments made by the reviewers, and our responses to them in a point-by-point format.

The only notable change made to the manuscript that was not in response to comments from a reviewer was regarding nomenclature of the structure that we had previously called the nuclear microtubule organising centre (MTOC). We had used the term MTOC to describe the entire structure, which spans the nuclear envelope and comprises an intranuclear portion and cytoplasmic extensions. Given recent evidence, including findings from this study, it is possible that both the intranuclear region and cytoplasmic extensions both have microtubule nucleating capacity, and therefore both meet the definition of an MTOC. To disambiguate this, we now refer to the overall structure as the centriolar plaque (CP), consistent with previous literature. The intranuclear portion of the CP will be referred to as the inner CP, while the cytoplasmic portion will be referred to as the outer CP.

**Reviewer #1 (Recommendations For The Authors):**
1. In the first part of the result section, a paragraph on sample processing for U-ExM could be added, with reference to Fig 1b.

The following section has been added to the first paragraph of the results “…In this study all parasites were fixed in 4% paraformaldehyde (PFA), unless otherwise stated, and anchored overnight at 37 °C before gelation, denaturation at 95 °C and expansion. Expanded gels were measured, before shrinking in PBS, antibody staining, washing, re-expansion, and imaging (Figure 1b). Parasites were harvested at multiple time points during the intraerythrocytic asexual stage and imaged using Airyscan2 super-resolution microscopy, providing high-resolution three-dimensional imaging data (Figure 1c). A full summary of all target-specific stains used in this study can be found in Figure 1d.”

1. The order of the figures could be changed for more consistency. For example, fig 2b is cited before 2a.

An earlier reference to figure 2a was added to rectify this discrepancy.

1. In Fig 2b it is difficult to distinguish the blue (nuclear) and green (plasma membrane) lines.x

The thickness of these lines has been doubled.

1. It is unclear what the authors want to show in Fig 2a.

The intention of this figure, as with panel a of the majority of the organelle-specific figures in this manuscript, is simply to show what the target protein/structure looks like across intraerythrocytic development.

1. Lines 154-155, the numbers of MTOC observed do not match those in Supplt Fig2c.

This discrepancy has been addressed, the numbers in Supplementary Figure 2c were accurate so the text has been changed to reflect this.

1. Line 188: the authors should explain the principle of C1 treatment.

The following explanation of C1 treatment has been provided:

“To ensure imaged parasites were fully segmented, we arrested parasite development by adding the reversible protein kinase G inhibitor Compound 1 (C1). This inhibitor arrests parasite maturation after the completion of segmentation but before egress. When C1 is washed out, parasites egress and invade normally, ensuring that observations made in C1-arrested parasites are physiologically relevant and not a developmental artefact due to arrest.”

1. Lines 195-204: this part is rather difficult to follow as analysis of the basal complex is detailed later in the manuscript. The authors refer to Fig4 before describing Fig3.

This has been clarified in the text

1. Lines 225 and 227, the authors cite Supplt Fig 2b about the Golgi, but probably meant Supplt Fig 4? In Supplt Fig 4, the authors could provide magnification in insets to better illustrate the Golgi-MTOC association.

This should have been a reference to Supplementary Figure 2e instead of 2b, which has now been changed. In Supplementary Figure 4, zooms into a single region of Golgi have been provided to more clearly show its MTOC association.

1. Supplt Fig8 is wrong (duplication of Supplt Fig6).

We apologise for this mistake, the correct figure is now present in Supplementary Figure 8.

1. Line 346: smV5 should be defined, and generation of the parasites should be described in the methods.

This has now been defined, but we have not described the generation of the parasites, as this was performed in a previous study that we have referenced.

1. Lines 361-362: "By the time the basal complex reaches its maximum diameter..." This sentence is not very clear, the authors could explain more precisely the sequence of events, indicating that the basal complex starts moving in the basal direction, as clearly illustrated in Fig 4a.

This has been prefaced with the following sentence “…As the parasite undergoes segmentation, the basal complex expands and starts moving in the basal direction.”

1. Supplt Fig6 comes after Supplt Fig9 in the narrative, and therefore could be placed after.

Supplementary Figure 6 and 9 follow the order in which they are referred to in the text.

1. Line 538: Supplt Fig9e instead of 9d.

This has been fixed.

1. Line 581: does the PFA-glutaraldehyde fixation allows visualizing other structures in addition to cytostome bulbs?

While PFA-glutaraldehyde fixation allows visualisation of cytostome bulbs, to date we have not observed any other structure that stains/preserves better using NHS Ester or BODIPY Ceramide in PFA-glutaraldehyde fixed parasites. As a general trend, all structures other than cytostomes become somewhat more difficult to identify using NHS Ester or BODIPY Ceramide in PFA-glutaraldehyde fixed samples due to the local contrast with the red blood cell cytoplasm. It seems likely that this is just due to the preservation of RBC cytoplasm, and would be expected from any fixation method that doesn’t result in RBC lysis, rather than anything unique to glutaraldehyde.

1. Line 652-653: It is unclear how the authors can hypothesize that rhoptries form de novo rather than splitting based on their observations.

This not something we can say with certainty, we have however, introduced the following paragraph to qualify our claims: “Overall, we present three main observations suggesting that rhoptry pairs undergo sequential de novo biogenesis rather than dividing from a single precursor rhoptry. First, the tight correlation between rhoptry and MTOC cytoplasmic extension number suggests that either rhoptry division happens so fast that transition states are not observable with these methods or that each rhoptry forms de novo and such transition states do not exist. Second, the heterogeneity in rhoptry size throughout schizogony favors a model of de novo biogenesis given that it would be unusual for a single rhoptry to divide into two rhoptries of different sizes. Lastly, well-documented heterogeneity in rhoptry density suggests that, at least during early segmentation, rhoptries have different compositions. Heterogeneity in rhoptry contents would be difficult to achieve so quickly after biogenesis if they formed through fission of a precursor rhoptry.”

1. Line 769: is expansion microscopy sample preparation compatible with FISH?

Yes, there are publications of expansion being done with both MERFISH and FISH. Though it has not yet been applied to plasmodium. See examples: Wang, Guiping, Jeffrey R. Moffitt, and Xiaowei Zhuang. "Multiplexed imaging of high-density libraries of RNAs with MERFISH and expansion microscopy." Scientific reports 8.1 (2018): 4847. And Chen, Fei, et al. "Nanoscale imaging of RNA with expansion microscopy." Nature methods 13.8 (2016): 679-684.

1. In the methods, the authors could provide details on the gel mounting step for imaging This is particularly important since this paper will likely serve as a reference standard for expansion microscopy in the field. Also, illustration that cryopreservation of gels does not modify the quality of the images would be useful.

The following section has been added to our “image acquisition” paragraph: “Immediately before imaging, a small slice of gel ~10mm x ~10mm was cut and mounted on an imaging dish (35mm Cellvis coverslip bottomed dishes NC0409658 - FisherScientific) coated with Poly-D lysine. The side of the gel containing sample is placed face down on the coverslip and a few drops of ddH20 are added after mounting to prevent gel shrinkage due to dehydration during imaging.”

We have decided not to illustrate that cryopreservation does not alter gel quality, as this is something that is already covered in the study that first cryopreserved gels, which is referenced in our methods section.

**Reviewer #2 (Recommendations For The Authors):**
1. Advantages and limitations of the expansion method are generally well discussed. The only matter in that respect that I was wondering is if expansion can always be assumed to be linear for all components of a cell. The hemozoin crystal does not expand (maybe not surprisingly), but could there also be other cellular structures that on a smaller scale separate or expand at a different rate than others? Is there any data on this from other organisms? I am raising this here not as a criticism of this work but if known to occur, it might need mentioning somewhere to alert the reader to it, particularly in regards to the many measurements in the paper (see also point 4). This might be a further factor contributing to the finding that the IMC and PPM could not be resolved.

This is an excellent point and, to our knowledge, one that is currently still under investigation in the field. It is well-documented that expansion protocols need to be customized to each cell type and tissue they are applied to. Each solution used for fixation and anchoring as well as timing and temperature of denaturation can affect the expansion factor achieved as well as how isotropic/anisotropic the expanded structures turn out. However, we do not know of any examples where isotropic expansion was achieved for everything but an organelle or component of the cell. It is our impression that if the cell seems to have attained isotropic expansion, this is assumed to also be the case for the subcellular structures within it. Nonetheless, we think it remains a possibility to be considered specially as more structures are characterized using these methods. In the case of our IMC/PPM findings, when we performed calculations taking into account our experimental expansion factor as well as antibody effects, it was clear that the resolution of our microscope was not enough to resolve the two structures using our current labelling methods. So, we suspect most of the effect is driven by that. However, this still needs to be validated by attempting to resolve the two structures though alternative labelling and imaging methods.

1. I understand that many things described in the results part are interconnected but still the level of hopping around between different figures/supp figures is considerable (see also point 6 on synchronicity of Figure parts). I do not have a simple fix, but maybe the authors could check if they could come up with a way to streamline parts of their results into a somewhat more reader friendly order.

This has been a problem we encountered from the beginning and, after trying multiple presentations of the results and discussion, we realized they all have drawbacks. We eventually settled on this presentation as the “least confusing”. We agree, however, that the figure references and order could be better streamlined and have addressed this to the best of our ability.

1. Are the authors sure the ER expands well and the BIP signal (Fig. S5) gives a signal reflecting the true shape of the ER? The signal in younger parasites seems rather extensive compared to what the ER (in my experience) typically looks like in these stages in live parasites.

While there may be a discrepancy between how the presumably dynamic ER appears in live cells, and how it appears using BiP staining, we think it is unlikely this is a product of expansion. Additionally, if there were to be an artefactual change in the ER, it would be likely under-expansion rather than over-expansion, which to our knowledge has not been reported. In our opinion, the BiP staining we observe is comparable between unexpanded and expanded samples. We have included comparative images in Author response image 1 with DNA in cyan and BiP in yellow, unexpanded (left) and expanded (right) using the same microscope and BiP antibody.

**Author response image 1. sa4fig1:** 

1. It is nice to have measurements of the apicoplast and mitochondria, but given their size, this could also have been done in unexpanded, ideally live parasites, avoiding expansion and fixing artifacts. While the expansion has many nice features, measuring area of large structures may not be one where it is strictly needed. I am not saying this is not useful information, but maybe a note could be added to the manuscript that the conclusions on mitochondria and apicoplast area and division might be worth confirming in live parasites. A brief mention on similarities and differences to previous work analysing the shape and multiplication of these organelles through blood stage development (van Dooren et al MolMicrobiol2005) might also be useful.

We agree with the reviewer that previous studies such as van Dooren et al. (2005) demonstrate that it is possible to track apicoplast and mitochondrial growth without expansion and share the opinion that live parasites are better for these measurements. Expansion only provides an advantage when more organelle-level resolution is needed. For example, in studying the association between these organelles and the MTOC or visualizing other branch-specific interactions.

1. I could not find the Supp Fig. 8 on the IMC, the current Supp Fig. 8 is a duplication of Supp Fig. 6

This has been addressed, Supplementary Figure 8 now refers to the IMC.

1. Figure order is not very synchronous with the text: Fig. 2a is mentioned after Fig. 2b, Fig. 4b is mentioned first for Fig. 4 (Fig. 4a is not by itself mentioned) and before Fig. 3 is mentioned; Fig. 3b is before Fig. 3a.

We have done our best to fix these discrepancies, but concede that we have not found a way to order these sections that doesn’t lead to some confusion.

1. Fig. S2a, The label "Centrin" on left image is difficult to read

We have increased the font size and changed colour slightly in the hope it is leigible.

1. In Fig. 2a, the centrin foci are very focal and difficult to see in these images, particularly when printed out but also on screen. To a lesser extent this is also the case for CINCH in Fig. 4a (particularly when printed; when zoomed-in on screen, the signal is well visible). This issue of difficulties in seeing the fluorescence signal of some markers, particularly when printed out, applies also to other images of the paper.

In the images of full size parasites, this is an issue that we cannot easily overcome as the fluorescent channels are already at maximum brightness without overexposure. To try and address this, we have provided zooms that we hope will more clearly show the fluorescence in these panels.

1. Expand "C1" in line 188 (first use).

This has been addressed in response to a previous comment.

1. Line 227; does Supp Fig. 2b really show Golgi- cytoplasmic MTOC association?

We have rephrased the wording of this section to clarify that we are observing proximity and not necessarily a physical tethering, however it is worth nothing that this was an accidental reference to Supplementary Figure 2b, and should’ve been Supplementary Figure 2e.

1. Line 230, in segmented schizonts the Golgi was considered to be at the apical end. It might be more precise to call its location to be close to the nucleus on the side facing the apical end of the parasite. It seems to me it often tends to be closer to the nucleus (in line with its proximity to the ER, see also point 13).

We have added more detail to this description clarifying that despite being at the apical end, the Golgi is closer to the nucleus.

1. Supp Fig. S5: Is the top cell indeed a ring? In the second cell there seem to be two nuclei, I assume this is a double infection (please indicate this in the legend or use images of a single infection).

In our opinion, the top cell in Supplementary Figure 5 is a ring. This is based on its size and its lack of an observable food vacuole (an area that lacks NHS ester staining). We typically showed images of ameoboid rings to avoid this ambiguity, but we think this parasite is a ring nonetheless. For the second image, this parasite is not doubly infected, as both DNA masses are actually contained within the same dumbbell shaped nuclear envelope. This parasite is likely undergoing its first anaphase (or the Plasmodium equivalent of anaphase) and will likely soon undergo its first nuclear division to separate these two DNA masses into individual nuclei.

1. Line 244: I would not call the Golgi a part of the apical cluster of organelles. All secretory cargo originates from the ER-Golgi-transGolgi axis in a directional manner and this axis is connected to the nucleus by the perinuclear ER. If seen from a secretory pathway centred view, it is the other way around and you could call the apical organelles part of the nuclear periphery which would be equally non-ideal.

Everything is close together in such a small cell. The secretory pathway likely is arranged in a serial manner starting from the perinuclear region to the transGolgi where cargo is sorted into vesicles for different destinations of which one is for the delivery of material to the apical organelles. The proposition that the Golgi is part of the apical cluster therefore somehow feels wrong, as the Golgi can still be considered to be upstream of the transGolgi before apical cargo branches off from other cargo destined for other destinationsWe agree with the reviewer that claiming a functional association between the Golgi and the apical organelles would be odd and we by no means meant to imply such functional grouping. Our intent was to confirm observations previously made about Golgi positioning by electron microscopy studies such as Bannister et al. (2000) at a larger spatial and temporal scale. These studies make the observation that the Golgi is spatially associated with the rhoptries at the apical end of the parasites. Logically, the Golgi is tied to the apical organelles through the secretory pathway as the reviewer suggests, but we claim no further relationship beyond that of organelle biogenesis. We have made modifications to the text to clarify these points.

1. Lines 300 - 308 (and thereafter): I assume these were also expanded parasites and the microtubule length is given after correction for expansion. I would recommend to indicate in line 274 (when first explaining the expansion factor) that all following measurements in the text represent corrected measures or, if this is not always the case, indicate on each occasion. Is the expansion factor accurate and homogenous enough to draw firm conclusions (see also point 1)? Could it be a reason for the variation seen with SPMTs? Could a cellular reference be used as a surrogate to account for cell specific expansion or would you assume that cellular substructure specific expansion differences exist and prevent this?

This is correct, the reported number is the number corrected for expansion factor, and the corresponding graphs with uncorrected data are present in the Supplementary Figures. We have clarified this in the text.Uneven expansion can be caused when certain organelles/structures do not properly denature. Given that out protocol denatures using highly concentrated SDS at 95 °C for 90 minutes, we do not anticipate that any subcellular compartments would expand significantly differently. In this study our expansion factors varied from ~4.1-4.7 across all gels, and for our corrected values we used the median expansion factor of 4.25. If we are interpreting the length of an interpolar spindle as 20 µm for example, the value would be corrected value would be 4.7 µm when divided by the median expansion factor, 4.9 µm when divided by the lowest, and 4.2 µm when divided by the highest. These values fall well within the measurement error, and so we expect that these small deviations in expansion factor between gels have a fairly minimal influence on variation in microtubule lengths.

1. Line 353: this is non-essential, but a 3D view of the broken basal ring might better illustrate the 2 semicircles

We have added the following panel to Supplementary Figure 3 to illustrate this more clearly:

**Author response image 2. sa4fig2:** 

1. The way the figure legends are shaped, it often seems only panel (a) is from expansion microscopy while the microscopy images in the other parts of the figures have no information on the method used. I assume all images are from expansion microscopy, maybe this could be clarified by placing this statement in a position of the legend that makes it clear it is for all images in a figure.

This has been clarified in the figure legends.

1. Fig. 8b, is it clear that internal RON4 is not below or above? Consider showing a 3D representation or side view of these max projections.

If in these images, we imagine we are looking at the ‘top’ of the rhoptries, our feeling is that the RON4 signal is on the ‘bottom’, at the part closest to the apical polar ring. We tried projecting this, however, but the images were not particularly due to spherical aberrations. Because of this, we have refrained from commenting on the RON4 location relative to the rhoptry bulb prior to elongation.

1. Line 684 "...distribution or RON4": replace or with of. The information of the next sentence is partly redundant, consider adding it in brackets.

This has been addressed.

1. Fig. 9a the EBA175 signal is not very prominent and a bit noisy, are the authors confident this is indeed showing only EBA175 or is there also some background?-AK

We agree with the reviewer that the EBA175 antibody shows a significant amount of background fluorescence, specially in the food vacuole area. However, we think the puncta corresponding to micronemal EBA175 can be clearly distinguished from background.

1. Fig. 9b, the long appearance of the micronemes in the z-dimension likely is due to axial stretch (due to point spread function in z and refractive index mismatch), in reality they probably are more spherical. It might be worth mentioning somewhere that this likely is not how these organelles are really shaped in that dimension (spherical fluorescent beads could give an estimation of that effect in the microscopy setup used).

After recently acquiring a water-immersion objective lens for comparison, it is clear that the transition from oil to hydrogel causes a degree of spherical aberration in the Z-plane, which in this instance causes the micronemes to be more oblong. As we make no conclusions based on the shape of the micronemes, however, we don’t think this is a significant consideration. This is an assumption that should be made when looking at any image whose resolution is not equal in all 3-dimensions. We also note that the more spherical shape of micronemes can be inferred from the max intensity projections in Figure 9c.

1. Fig. 9b, the authors mention in the text that there is NHS ester signal that overlaps with the fluorescence signal, can occasions of this be indicated in the figure?

Figure 9b was already quite busy, so we instead added the following extra panel to this figure that more clearly shows the NHS punctae we thought may have been micronemes:

**Author response image 3. sa4fig3:** 

1. Fig. 9, line 695, the authors write that the EBA puncta were the same size as AMA1 puncta. To me it seems the AMA1 areas are larger than the EBA foci, is their size indeed similar? Was this measured?

Since we did not conduct any measurements and doing so robustly would be difficult given the density of the puncta, we have decided to remove our comment on the relative size of the puncta.

1. Materials and methods: Remove "to" in line 871; explain bicarb and incomplete medium in line 885 (non-malaria researchers will not understand what is meant here); line 911 and start of 912 seem somewhat redundant

This has been addressed.

1. is there more information on what the Airyscan processing at moderate filter level does? The background of the images seems to have an intensity of 0 which in standard microscopy images should be avoided (see for instance doi:10.1242/jcs.03433) similar to the general standard of avoiding entirely white backgrounds on Western blots. I understand that some background subtraction processes will legitimately result in this but then it would be nice to know a bit better what happened to the original image.

We have taken the following excerpt from a publication on Airyscan to help clarify:

"Airyscan processing consists of deconvolution and pixel reassignment, which yield an image with higher resolution and reduced noise. This can be a contributor to the low background in some channels. The level of filtering is the processing strength, with higher filtering giving higher resolution but increased chances of artefacts. More information about the principles behind Airyscan processing can be found in the following two publications, though details on the algorithm itself seem to be proprietary: Huff, Joseph. "The Airyscan detector from ZEISS: confocal imaging with improved signal-to-noise ratio and super-resolution." (2015): i-ii. AND Wu, Xufeng, and John A. Hammer. "Zeiss airyscan: Optimizing usage for fast, gentle, super-resolution imaging." Confocal Microscopy: Methods and Protocols. New York, NY: Springer US, 2021. 111-130."

We cannot find any further information about the specifics of Airyscan filtering, however, the moderate filter that we used is the default setting. This information was included just for clarity, rather than something we determined by comparison to other filtering settings.

In regards to the background, the majority of some images having an intensity value of 0 is partially out of our control. For all NHS Ester images, the black point of the images was 0 so areas that lack signal (white in the case of NHS Ester) truly had no signal detected for those pixels. While we appreciate that never altering the black point of images displays 100% of the data in the image, images with any significant background can become impossibly difficult to interpret. We have done our best to try and present images where the black point is modified to remove background for ease of interpretation by the readers only.

**Reviewer #3 (Public Review):**
1. Most importantly, in order to justify the authors claim to provide an "Atlas", I want to strongly suggest they share their raw 3D-imaging data (at least of the main figures) in a data repository. This would allow the readers to browse their structure of interest in 3D and significantly improve the impact of their study in the malaria cell biology field.

We agree completely that the potential impact of this study is magnified by public sharing of the data. The reason that this was not done at the time of submission is that most public repositories do not allow continued deposition of data, and so new images included in response to reviewers comments would’ve been separated from the initial submission, which we saw as needlessly complicated. All 647 images that underpin the results discussed in this manuscript are now publicly available in Dryad (https://doi.org/10.5061/dryad.9s4mw6mp4)

1. The organization of the manuscript can be improved. Aside some obvious modifications as citing the figures in the correct order (see also further comments and recommendations), I would maybe suggest one subsection and one figure per analyzed cellular structure/organelle (i.e. 13 sections). This would in my opinion improve readability and facilitate "browsing the atlas".

This is actually how we had originally formatted this manuscript, but this structure made discussing inter-connected organelles, such as the IMC and basal complex, impossibly difficult to navigate. We have done our best to make the manuscript flow better, but have not come up with any way to greatly restructure the manuscript so to increase its readability.

1. Considering the importance of reliability of the U-ExM protocol for this study the authors should provide some validation for the isotropic expansion of the sample e.g. by measuring one well defined cellular structure.

The protocol we used comes from the Bertiaux et al., 2021 PLoS Biology study. In this study they show isotropic expansion of blood-stage parasites.

1. In the absence of time-resolved data and more in-depth mechanistic analysis the authors must down tone some of their conclusions specifically around mitochondrial membrane potential, subpellicular microtubule depolymerization, and kinetics of the basal complex.

Our conclusions regarding mitochondrial membrane potential and basal complex kinetics have been dampened. We have not, however, changed our wording around microtubule depolymerisation. Partial depolymerisation of microtubules during fixation is a known phenomenon in Plasmodium, and in our opinion, our explanation of this offers a hypothesis that is balanced with respective to evidence: “we hypothesise that most SPMTs measured in our C1-treated schizonts had partially depolymerised. *P. falciparum* microtubules are known to rapidly depolymerise during fixation10,29. It is unclear, however, why this depolymerization was observed most often in C1-arrested parasites. Thus, we cannot determine whether these shorter microtubules are a by-product of drug-induced arrest or a biologically relevant native state that occurs at the end of segmentation.”

1. The observation that the centriolar plaque extensions remains consistently tethered to the plasma membrane is of high significance. To more convincingly demonstrate this point, it would be very helpful to show one zoomed-in side view of nucleus with a mitotic spindle were both centriolar plaques are in contact with the plasma membrane.

We of course agree that this is one of our most important observations, but in our opinion this is already demonstrated in Figure 2b. The third panel from the right shows a mitotic spindle and has the location of the cytoplasmic extensions, nuclear envelope and parasite plasma membranes annotated.

1. Please verify the consistent use of the term trophozoite and schizont. In Fig. 1c a parasite with two nuclei, likely in the process of karyofission is designated as trophozoite, which contrasts with the mononucleated trophozoite shown in Fig. 1a. The reviewer is aware of the more "classical" description of the schizont as parasite with more than 2 nuclei, but based on the authors advanced knowledge of cell cycle progression and mitosis I would encourage them to make a clear distinction between parasites that have entered mitotic stages and pre-mitotic parasites (e.g. by applying the term schizont, and trophozoite, respectively).

For this study, we have interpreted any parasite having three or more nuclei as being a schizont. We are aware this morphological interpretation is not universally held and indeed suboptimal for studying some aspects of parasite development, but all definitions of a schizont have some drawbacks. Whether a parasite has entered mitosis or not is obviously a hugely significant event in the context of cell biology, but in a mononucleated parasite this could only be determined using immunofluorescence microscopy with cell cycle or DNA replication markers.

1. Aldolase does not localize diffusely in the cytoplasm in schizont stages as in contrast to earlier stage. The authors should comment on that.

We are unclear if this is an interpretation of the images in supplementary figure 1, or inferred from other studies. If this is an interpretation of the images in Supplementary Figure 1, we do not agree that the images show a significant change in the localisation of aldolase. It is possible that this difference in interpretation comes from the strong punctate signal observed more readily in the schizont images. This is the strong background signal in or around the food vacuole we mention in the text. These punctae are significantly brighter than the cytosolic aldolase signal, making it difficult to see them on the aldolase only channel, but aldolase signal can clearly be seen in the cytoplasm on the merge images.

1. Line 79. Uranyl acetate is just one of the contrasting agents used in electron microscopy. The authors might reformulate this statement. Possibly this would also be a good opportunity to briefly mention that electron density measured in EM and protein-density labeled by NHS-Ester can be similar but are not equivalent.

We have expanded on this in the text.

1. The authors claim that they investigate the association between the MTOC and the APR (line 194), but strictly speaking only look at subpellicular microtubules and an associated protein density. The argument that there is a "NHS ester-dense focus" (line 210) without actual APR marker is not quite convincing enough to definitively designate this as the APR.

While an APR marker would of course be very useful, there are currently no published examples of APR markers in blood-stage parasites. We therefore think that the timing of appearance, location, and staining density are sufficient for identifying this structure as the APR, as it has previously been designated through EM studies. We have nonetheless softened our language around APR-related observations.

1. Line 226: The authors should also discuss the organization of the Golgi in early schizonts (Fig. S4). (not only 2 nuclei and segmenter stages).

We did not mean to imply that all 22 parasites had only 2 nuclei, but instead that they had 2 or more nuclei. Therefore, early schizonts are included in this analysis, with Golgi closely associated with all their MTOCs.

1. Line 242: To the knowledge of the reviewer the nuclear pore complexes, although clustered in merozoites and ring stages, don't particularly "define the apical end of the parasite".

The MTOC is surrounded by NPCs, which because of the location of the MTOC end up being near the forming apical end of the merozoite, but we have removed this as it was needlessly confusing.

1. Supplementary Figure 8 is missing (it's a repetition of Fig. S6).

This has been addressed.

1. Line 253: asexual blood stage parasites have two classes of MTs. Other stages can have more.

This has been clarified.

1. Fig. 3f: Please comment how much of these observations of "only one" SPMT could result from suboptimal resolution (e.g. in z-direction) or labeling. Otherwise use line profiles to argue that you can always safely distinguish SPMT pairs.

In the small number of electron tomograms of merozoites where the subpellicular microtubules have been rendered, they have been seen to have 2 or 3 SPMTs. Despite this, we don’t think it is likely that the single SPMT merozoites observed in this study are caused by a resolution limitation. SPMTs were measured in 3D, rather than from projections, and any schizont where the SPMTs were pointing towards the objective lens, elongating the parasite in Z, were not imaged. Additionally, our number of merozoites with a single SPMT correspond with the same data collected in the Bertiaux et al., 2021 PLoS Biology study. We cannot rule this out as a possibility, as sometimes SPMTs cross over each other in three-dimensions, and at these intersection points they cannot be individually resolved. We, however, think it is very unlikely that two SPMTs would be so close that they can never be resolved across any part of their length.

1. Lines 302ff: the claim that variability in SPMT size must be a consequence of depolymerzation is unfounded. The dynamics of SPMT are unknown at this point. Similarly unfounded is the definitive claim that it is known that P.f. MTs depolymerize upon fixation. Other possibilities should be considered. SPMT could also simply shorten in C1-arrested parasites.

While we agree with the reviewer that much about SPMT dynamics in schizonts remains unknown, we disagree with the claim that our consideration of SPMT depolymerization as a possible explanation for our observations is unfounded. Microtubule depolymerization is a well-known fixation and sample preparation artefact in both mammalian cells and a well-documented phenomenon in Plasmodium when parasites are washed with PBS prior to fixation. We convey in the text our belief that it is possible that SPMTs shorten in C1-arrested parasites as a result of drug treatment. However, it is our opinion that there simply is not enough evidence at this moment to conclusively pinpoint the cause of our observed depolymerization. As we mention in the text, further experiments are needed in order to determine with confidence whether depolymerization is a consequence of our fixation protocol, a consequence of C1 treatment (or the length of that treatment), or a biological phenomenon resulting from parasite maturation.

1. Line 324: "up to 30 daughter merozoites"

Schizonts can have more than 30 daughter merozoites, so we have not altered this statement.

1. Figure 4b. Line 354 The postulated breaking in two is not well visible and here the authors should attempt a more conservative interpretation of the data (especially with respect to those early basal complex dynamics).

We think that the basal complex dividing or breaking in two is the more conservative interpretation of our data. There is no evidence to suggest that a second basal complex is formed de novo and, while never before described using a basal complex protein, the cramp-like structure and dynamics we observe are consistent with that observed in early IMC proteins. We have updated the text to provide additional context and make the reasoning behind our hypothesis clearer.

1. Line 365: Commenting on their relative size would require a quantification of APR and basal complex size (can be provided in the text).

We are unsure what this is in reference to, as there is no mention of the APR in the basal complex section.

1. Lines 375ff: The claim that NHS Ester is a basal complex marker should be mitigated or more convincing images without the context of anti-CINCH staining being sufficient to identify the ring structure should be presented.

We have provided high quality, zoomed-in images without anti-CINCH staining in Fig. 5D&E, 6C, 7b, and Supplementary Fig. 8 that show that even in the absence of a basal complex antibody, the basal complex still stains densely by NHS ester.

.20. Line 407: The claim that there are differences in membrane potential along the mitochondria needs to be significantly mitigated. There are several alternative explanations of this staining pattern (some of which the authors name themselves). Differences in local compartment volume, differences in membrane surface, diffusibility/leakage of the dye can definitively play a role in addition to fixation and staining artefacts (also brought forward recently for U-ExM by Laporte et al. 2022 Nat Meth). Confirming the hypothesis of the authors would need significantly more experimental evidence that is outside the scope of this study.

We have significantly dampened and qualified the wording in this section. It now reads: “These clustered areas of MitoTracker staining were highly heterogeneous in size and pattern. Small staining discontinuities like these are commonly observed in mammalian cells when using MitoTracker dyes due to the heterogeneity of membrane potential from cristae to cristae as well as due to fixation artifacts. At this point, we cannot determine whether the staining we observed represents a true biological phenomenon or an artefact of this sample preparation approach. Our observed MitoTracker-enriched pockets could be an artifact of PFA fixation, a product of local membrane depolarization, a consequence of heterogeneous dye retention, or a product of irregular compartments of high membrane potential within the mitochondrion, to mention a few possibilities. Further research is needed to conclusively pinpoint an explanation.”

1. Fig. 7e: The differences in morphology using different fixation methods are interesting. Can the authors provide a co-staining of K13-GFP together with the better-preserved structures in the GA-containing fixation protocol to demonstrate that these are indeed cytostome bulbs?

Figure 7 has been changed substantially to show more clearly the preservation of the red blood cell membrane following PFA-GA fixation, followed by direct comparison of K13-GFP stained parasites fixed in either PFA only or PFA-GA. The cytostome section of the results has also changed to reflect this, the changed section now reads:

“PFA-glutaraldehyde fixation allows visualization of cytostome bulbThe cytostome can be divided into two main components: the collar, a protein dense ring at the parasite plasma membrane where K13 is located, and the bulb, a membrane invagination containing red blood cell cytoplasm {Milani, 2015 #63;Xie, 2020 #62}.While we could identify the cytostomal collar by K13 staining, these cytostomal collars were not attached to a membranous invagination. Fixation using 4% v/v paraformaldehyde (PFA) is known to result in the permeabilization of the RBC membrane and loss of its cytoplasmic contents65. Topologically, the cytostome is contiguous with the RBC cytoplasm and so we hypothesised that PFA fixation was resulting in the loss of cytostomal contents and obscuring of the bulb. PFA-glutaraldehyde fixation has been shown to better preserve the RBC cytoplasm65. Comparing PFA only with PFA-glutaraldehyde fixed parasites, we could clearly observe that the addition of glutaraldehyde preserves both the RBC membrane and RBC cytoplasmic contents (Figure 7c). Further, while only cytostomal collars could be observed with PFA only fixation, large membrane invaginations (cytostomal bulbs) were observed with PFA-glutaraldehyde fixation (Figure 7d). Cytostomal bulbs were often much longer and more elaborate spreading through much of the parasite (Supplementary Video 1), but these images are visually complex and difficult to project so images displayed in Figure 7 show relatively smaller cytostomal bulbs. Collectively, this data supports the hypothesis that these NHS-ester-dense rings are indeed cytostomes and that endocytosis can be studied using U-ExM, but PFA-glutaraldehyde fixation is required to maintain cytostome bulb integrity.”

1. It would be helpful to the readers to indicate in the schematic in Fig. 1b at which point NHS-Ester staining is implemented.

Figure 1b is slightly simplified in the sense that it doesn’t differentiate primary and secondary antibody staining, but we have updated it to reflect that antibody and dye staining are concurrent, rather than separate.

1. In Fig. 2B the second panel from the right the nuclear envelope boundary does not seem to be accurately draw as it includes the centrin signal of the centriolar plaque.

Thank you for pointing this out, it has now been redrawn.

1. Line 44-45: should read "up to 30 new daughter merozoites" (include citations).

We have included a citation here, but left it as approximately 30 daughter merozoites as the study found multiple cells with >30 daughter merozoites.

1. Line 49: considering its discovery in 2015 the statement that it has gained popularity in the last decade can probably be omitted.

This has been removed.

1. Fig S1 should probably read "2N" (instead of "2n"). Or alternatively "2C" could be fine.1. Line 154: To help comprehension please define the term "branch number" in this context when it comes up.

A definition for branch has now been provided.

1. Fig. S5: To my estimation it is not an "early trophozoite", which is depicted.

While this parasite technically fits our definition of trophozoite, as it has not yet undergone nuclear division, we have swapped it for a visibly earlier parasite for clarity. This is the new parasite depicted

**Author response image 4. sa4fig4:** 

1. Fig. 2a is not referenced before Fig. 2b in the text.

This has been addressed.

1. I could not find the reference to Fig. S2e and its discussion.

It was wrongly labelled as Supplementary Figure 2b in the text, this has now been addressed.

1. The next Figure referenced in the text after Fig. 2b is Fig. 4b. Fig.3 is only referenced and discussed later, which was quite confusing.

The numbering discrepancies have been addressed.

1. Line 196: Figure reference is missing.

This data did not have a figure reference, but the numbers have now been provided in-text.

1. Fig. 3c: Is "Branches per MTOC" not just total branches divided by two? If so it can be omitted. If not so please explain the difference.

Yes it was total branched divided by two, this has been removed from Figure 3c.

1. Figure 5c and 6d: The authors should show examples of the image segmentation used to calculate the surface area.

Surface area calculation was done in an essentially one step process. From maximum intensity projections, free-hand regions of interest were drawn, from which ZEN automatically calculates their area. Example as Author response image 5:

**Author response image 5. sa4fig5:** 

1. Figure 7b should also show the NHS Ester staining alone for the zoom in.

We have included the NHS ester staining alone on the zoom on, but we have slightly changed the presentation of these two panels to show both the basal complex and cytostomes as follows:

**Author response image 6. sa4fig6:** 

1. To which degree are Rhoptry necks associated with MTOC extensions?

This cannot easily be determined with the images we have so far. Before elongated necks are visible, the RON4 signal does appear pointed towards the MTOC extensions. Rhoptry necks don’t seem to elongate until segmentation, when the MTOC starts to move away from the apical end of the parasite. So it is possible there is a transient association, but we cannot easily discern this from our data.